# Lipophilic nanocrystal prodrug-release defines the extended pharmacokinetic profiles of a year-long cabotegravir

Nagsen Gautam[1], JoEllyn M. McMillan[2], Devendra Kumar[1], Aditya N. Bade[2], Qiaoyu Pan[1], Tanmay A. Kulkarni[1], Wenkuan Li[1], Brady Sillman [2], Nathan A. Smith[2], Bhagya L. Dyavar Shetty[2], Adam Szlachetka[3], Benson J. Edagwa [2], Howard E. Gendelman [1,2,4✉] & Yazen Alnouti[1,4✉]

A once every eight-week cabotegravir (CAB) long-acting parenteral is more effective than daily oral emtricitabine and tenofovir disoproxil fumarate in preventing human immunodeficiency virus type one (HIV-1) transmission. Extending CAB dosing to a yearly injectable advances efforts for the elimination of viral transmission. Here we report rigor, reproducibility and mechanistic insights for a year-long CAB injectable. Pharmacokinetic (PK) profiles of this nanoformulated CAB prodrug (NM2CAB) are affirmed at three independent research laboratories. PK profiles in mice and rats show plasma CAB levels at or above the protein-adjusted 90% inhibitory concentration for a year after a single dose. Sustained native and prodrug concentrations are at the muscle injection site and in lymphoid tissues. The results parallel NM2CAB uptake and retention in human macrophages. NM2CAB nanocrystals are stable in blood and tissue homogenates. The long apparent drug half-life follows pH-dependent prodrug hydrolysis upon slow prodrug nanocrystal dissolution and absorption. In contrast, solubilized prodrug is hydrolyzed in hours in plasma and tissues from multiple mammalian species. No toxicities are observed in animals. These results affirm the pharmacological properties and extended apparent half-life for a nanoformulated CAB prodrug. The report serves to support the mechanistic design for drug formulation safety, rigor and reproducibility.

[1] Department of Pharmaceutical Sciences, University of Nebraska Medical Center, Omaha, NE, USA. [2] Department of Pharmacology and Experimental Neuroscience, University of Nebraska Medical Center, Omaha, NE, USA. [3] Nebraska Nanomedicine Production Plant, University of Nebraska Medical Center, Omaha, NE, USA. [4] These authors jointly supervised this work: Howard E. Gendelman, Yazen Alnouti. ✉email: hegendel@unmc.edu; yalnouti@unmc.edu

While wide-spread availability of antiretroviral therapy (ART) has reduced human immunodeficiency virus type one (HIV-1) morbidities and mortality, viral transmission continues to persist. This is highlighted by yearly recordings of two million new infections worldwide[1]. ART requires life-long daily oral regimens with both short and long-term toxicities. Changing dosing regimens driven by the emergence of resistant viral strains and regimen adherence are therapeutic limitations[2–5]. A priority for HIV-1/acquired immune deficiency syndrome (AIDS) research is how best to prevent the spread of infection which can be achieved through ART-mandated pre-exposure prophylaxis (PrEP)[6]. PrEP success has been buoyed by a long-acting (LA) parenteral formulation of cabotegravir (CAB). This permits every other month antiretroviral drug (ARV) dosing. CAB LA was recently approved by the US Food and Drug Administration demonstrating improved pharmacokinetic and pharmacodynamic (PK and PD) profiles compared to more conventional ARV regimens and as such provides a promising strategy for HIV-1 prevention and treatment[7].

CAB is unique in its structural and antiretroviral properties. CAB LA is the first LA injectable regimen amongst the group of integrase strand transfer inhibitors (INSTI). These relatively new ARVs have complemented existing ARVs based on their unique treatment and improved PK profiles. Notably, CAB LA PK results are highlighted by an extended plasma half-life of up to 54 days[8] and limited drug–drug interactions (DDI). The latter is linked both to its membrane permeability and low affinity for cytochrome P450 (CYP450)[9,10]. Moreover, CAB has low aqueous solubility and a high melting point allowing its administration as a high concentration parenteral formulation[11]. CAB LA is currently completing phase 3 clinical trials and was approved recently for use in Canada. USA Food and Drug Administration (FDA) approval is anticipated in late spring-summer of 2021[12]. Nonetheless, there are limitations for CAB LA as the current formulation requires a 2 ml dosing volume known to elicit injection site reactions and continuous health care oversights[13]. Thus, newer formulations with longer dosing intervals, ease of access, and reduced administration volumes will positively impact wide-spread regimen use.

In order to overcome the treatment limitations of CAB LA we developed lipophilic fatty acid ester CAB prodrugs nanoformulations (named NMCAB and NM2CAB, catalogued based on carbon lengths) with extended PK properties. The PK profiles were improved from months (NMCAB)[14] to one year (NM2CAB)[15]. However, during the development of a year-long CAB, PK data set reproducibility in different animal species proved mandatory for development. This is boosted by unraveling the mechanism(s) of the extended drug apparent half-life that includes prodrug hydrolysis. Moreover, safety measurements for the year-long parenteral formulation were required. With these in mind, we now report, from separate laboratories, PK and biodistribution (BD) profiles, of NM2CAB, our lead nanoformulated stearoylated prodrug. Work was completed in rodents with head-to-head comparisons against a CAB LA formulation (reflected by NCAB) analogous to that currently being evaluated in human trials. Rigor and reproducibility are defined by PK studies by two research (from the University of Nebraska Medical Center Colleges of Medicine and Pharmacy) and one contract laboratory (Covance Laboratories, a global contract research organization and drug development services company) employing multiple mouse strains (immune deficient and competent) and rats. Lastly, mechanisms were elucidated together with the nanoformulation stability demonstrating that particle stability and slow prodrug release best reflected the extended drug half-life for the prodrug formulation rather than the prodrug enzymatic or chemical stability. Each of these tests proved critical in defining the safety and

reproducibility of a year-long CAB apparent half-life. The solid state form of M2CAB nanocrystals is a key determinant for prodrug stability in biological fluids and tissues. We posit that the translation of a safe biocompatible formulation will have a significant impact on the therapeutic armamentarium in preventing HIV/AIDS infection and in providing improved access to those populations most impacted by high viral transmission rates.

## Results

**Rigor and reproducibility.** The experiment results described were performed to ensure rigor into an extended LA concept for ARV delivery[15]. The concept centered on a transformation in drug apparent half-life made in enabling parenteral drug administration from a once a month or once every other month into a once-a-year drug administration regimen. The extension in drug usage was developed to ease adherence to PrEP treatments. The importance of the extended LA CAB regimen serves as a means to reduce transmission rates for HIV/AIDS affecting, most notably, resource-limited settings. Thus, the results outlined below were conducted to first affirm then extend prior published data. These data serve to substantiate the data sets in multiple species then defining the mechanisms for the extended half-life of the nanoformulated CAB prodrug[15].

In the first phase of these experiments, we evaluated the drug and prodrug PK and BD profiles in male Balb/cJ mice and Sprague Dawley (SD) rats that were injected intramuscularly (IM) with a single NCAB or NM2CAB dose of 45 mg CAB equivalents/kg. In mice, the peak CAB plasma concentration ($C_{max} = 49,066$ ng/ml) after NCAB administration was 6.5-fold higher than that of NM2CAB ($C_{max} = 7480$ ng/ml) and occurred after 24 h ($T_{max}$) for both formulations (Table 1). Following these recorded drug levels, CAB blood concentrations fell sharply for NCAB compared to NM2CAB. CAB terminal half-life ($t_{0.5}$) following NM2CAB (118.4 days) treatment was 16-fold greater than NCAB (7.5 days). Similarly, CAB mean residence time (MRT) after NM2CAB administration was 19-fold longer than that of NCAB (164.7 vs. 8.7 days, respectively). The longer CAB $t_{0.5}$ associated with NM2CAB is the result of a 20-fold higher apparent volume of distribution (Vz = 21.7 L/kg vs. 1.08 L/kg for NCAB). Starting at day 28 after NM2CAB administration, CAB plasma concentrations exceeded that following NCAB treatment at all time points, and by day 364 it was 280 times higher (263.2 ng/ml from NM2CAB vs. 0.9 ng/ml from NCAB). CAB plasma concentrations after NM2CAB administration remained higher than the CAB protein-adjusted 90% inhibitory concentration (PA-IC$_{90}$) of 166 ng/ml for the entire period of the study, compared to <42 days following NCAB administration (Fig. 1a). M2CAB prodrug concentrations in plasma and blood (Supplementary Fig. 1a) after day 1 were more than 100 times lower than CAB concentrations or undetectable after NM2CAB administration. The peak plasma concentration of M2CAB ($C_{max} = 558$ ng/ml) was detected immediately after NM2CAB administration at 4 h ($T_{max}$). After that, M2CAB concentrations declined rapidly and were undetected after day 42.

In rats, the peak plasma concentration ($C_{max} = 77,700$ ng/ml) after NCAB administration was 5.5 times higher than that of NM2CAB ($C_{max} = 14,123$ ng/ml) and occurred after 7 and 14 days ($T_{max}$) for NCAB and NM2CAB, respectively (Table 1). Similar to mice, CAB plasma concentrations fell sharply and at a much faster rate after NCAB administration compared to NM2CAB. CAB $t_{0.5}$ following NM2CAB (81.5 days) treatment was 5.2-fold greater than NCAB (15.7 days). Similarly, CAB MRT of NM2CAB was 5.5-fold longer than NCAB (117.7 vs. 21.6 days, respectively). The longer CAB $t_{0.5}$ associated with NM2CAB is also the result of a 6.2-fold higher apparent volume of distribution (Vz = 3.42 L/kg vs. 0.55 L/kg for NCAB). Starting

**Table 1 CAB PK parameters after NCAB and NM2CAB administrations.**

| PK parameters | Mouse | | Rat | | Monkey |
|---|---|---|---|---|---|
| | NCAB | NM2CAB | NCAB | NM2CAB | NM2CAB |
| $T_{max}$ (day) | 1.0 | 1.0 | 7.0 | 14.0 | 1.0 |
| $C_{max}$ (ng/ml) | 49,066.7 | 7480.0 | 77,700.0 | 14,123.3 | 2753.0 |
| $\lambda_Z$ (1/h) | 0.0927 | 0.0059 | 0.0443 | 0.0085 | 0.0038 |
| $t_{0.5}$ (day) | 7.5 | 118.4 | 15.7 | 81.5 | 184.5 |
| $AUC_{last}$ (day*ng/ml) | 450,906.3 | 309,193.1 | 1,850,671.2 | 1,466,869.0 | 74,955.9 |
| $AUC_{0-\infty}$ (day*ng/ml) | 450,916.4 | 354,157.1 | 1,850,738.1 | 1,545,319.2 | 79,919.1 |
| AUC_% Extrapolation | 0.2 | 12.70 | 0.004 | 5.08 | 6.21 |
| $V_Z/F$ (L/kg) | 1.08 | 21.70 | 0.55 | 3.42 | 149.84 |
| CL/F (L/day/kg) | 0.100 | 0.127 | 0.024 | 0.029 | 0.563 |
| MRT $0-\infty$ (day) | 8.8 | 164.8 | 21.6 | 117.7 | 228.4 |

PK parameters of CAB in Balb/cJ mice and Sprague-Dawley (SD) rats after NCAB and NM2CAB administration and in rhesus macaques after NM2CAB administration. PK parameters were calculated by non-compartmental analysis from the mean of $N = 6$ mice and rats, and $N = 4$ monkeys.

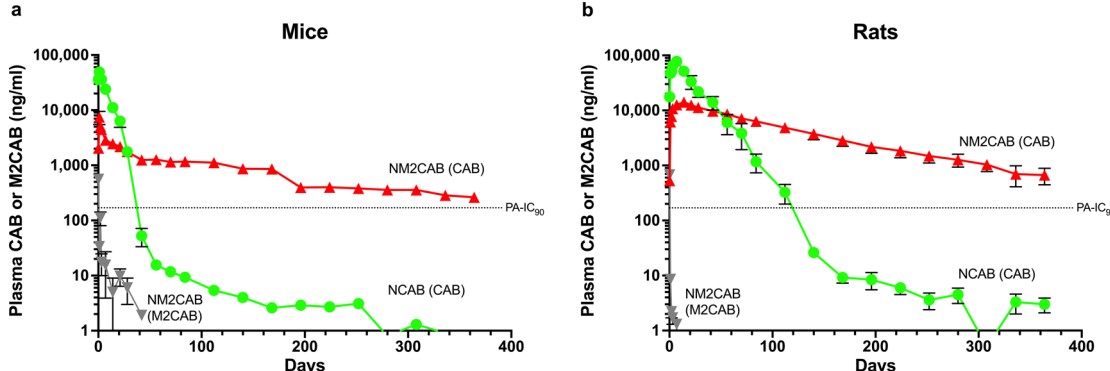

**Fig. 1 Plasma concentration vs. time profiles of CAB and M2CAB plasma concentrations in mice and rats over one year.** Plasma concentration over time was recorded for CAB and M2CAB over one year after a single IM injection (45 mg CAB equivalent/kg) of NCAB or NM2CAB in male Balb/cJ mice (**a**) and male SD rats (**b**). The dotted line indicates the CAB protein-adjusted $IC_{90}$ (PA-$IC_{90}$). For the NM2CAB treatment both the prodrug (M2CAB) and parent (CAB) were measured. Data are expressed as mean ± SEM ($N = 6$ animals per group per time point).

at day 56 after NM2CAB administration, CAB plasma concentrations exceeded that of NCAB at all-time points, and by the end of the study (day 364) were 225-fold higher (667.6 ng/ml from NM2CAB vs. 3.0 ng/ml from NCAB). CAB plasma concentrations after NM2CAB administration remained higher than the CAB PA-$IC_{90}$ (166 ng/ml) for the entire period of the study, compared to <140 days after NCAB administration (Fig. 1b). The peak plasma concentration of M2CAB ($C_{max} = 677$ ng/ml) was detected immediately after NM2CAB administration at 4 h ($T_{max}$). After that, M2CAB concentrations in plasma and blood (Supplementary Fig. 1b) sharply decreased and were undetected after day 7. In follow-on we extended the PK analyses in rhesus macaques from what was previously published for NM2CAB[15]. In these extended studies the PK profiles of NM2CAB were evaluated for up to 2 years. Animals were injected IM with a single dose of 45 mg CAB equivalents/kg of NM2CAB. Similar to mice and rats, NM2CAB was slowly eliminated from plasma, and CAB was detected for the entire period of the study. After that, CAB plasma concentrations decreased slowly over the period of ~2 years ($t_{0.5} = 184$ days and MRT = 228 days) (Table 1). No adverse responses were observed in any species tested. Rat serum metabolic panels at days 84 and 365 were similar to untreated controls following both NCAB and NM2CAB treatments (Supplementary Table 1). Our previous studies have shown normal metabolic panels in mice at one year following NM2CAB treatment[15]. In mice, the initial weight at dosing time was 18–23

g, and at the one-year study end was 28–35 g with no differences between treatments and controls. In rats, the initial weight at dosing time was 186–223 g, and by the one-year study end was 500–562 g with no differences between treatments and controls. No abnormalities were observed visually at the site of injection. Nonhuman primates (NHP) were monitored for up to two years with no evidence of any adverse reactions. Former studies provided one year only toxicological measures[15].

We sought to confirm these PK test results in studies conducted by an independent facility. For these experiments blinded samples of NCAB and NM2CAB formulations were sent to Covance, a contract research laboratory. These experiments were designed to provide rigor and reproducibility of the enhanced PK profiles for NM2CAB. PK tests were performed in two mouse strains, male and female, for NCAB and NM2CAB at two CAB equivalent doses for each formulation. Results from both mouse studies at doses of 45 or 70 mg CAB equivalents/kg showed that CAB plasma concentrations after NCAB administration were initially higher compared to NM2CAB (Fig. 2a–d). After 28 days, CAB concentrations fell sharply and at a much faster rate after NCAB compared to NM2CAB treatment after either dose. In both strains of mice given the lower dose of NCAB, CAB was below 10 ng/ml at three months, and at the limit of detection (1 ng/ml) after four months (Fig. 2a, b). In contrast, after NM2CAB administration CAB concentration decreased at a much slower rate and at 6 months was 437 and 575 ng/ml in male Balb/cJ and female NSG mice, respectively. The

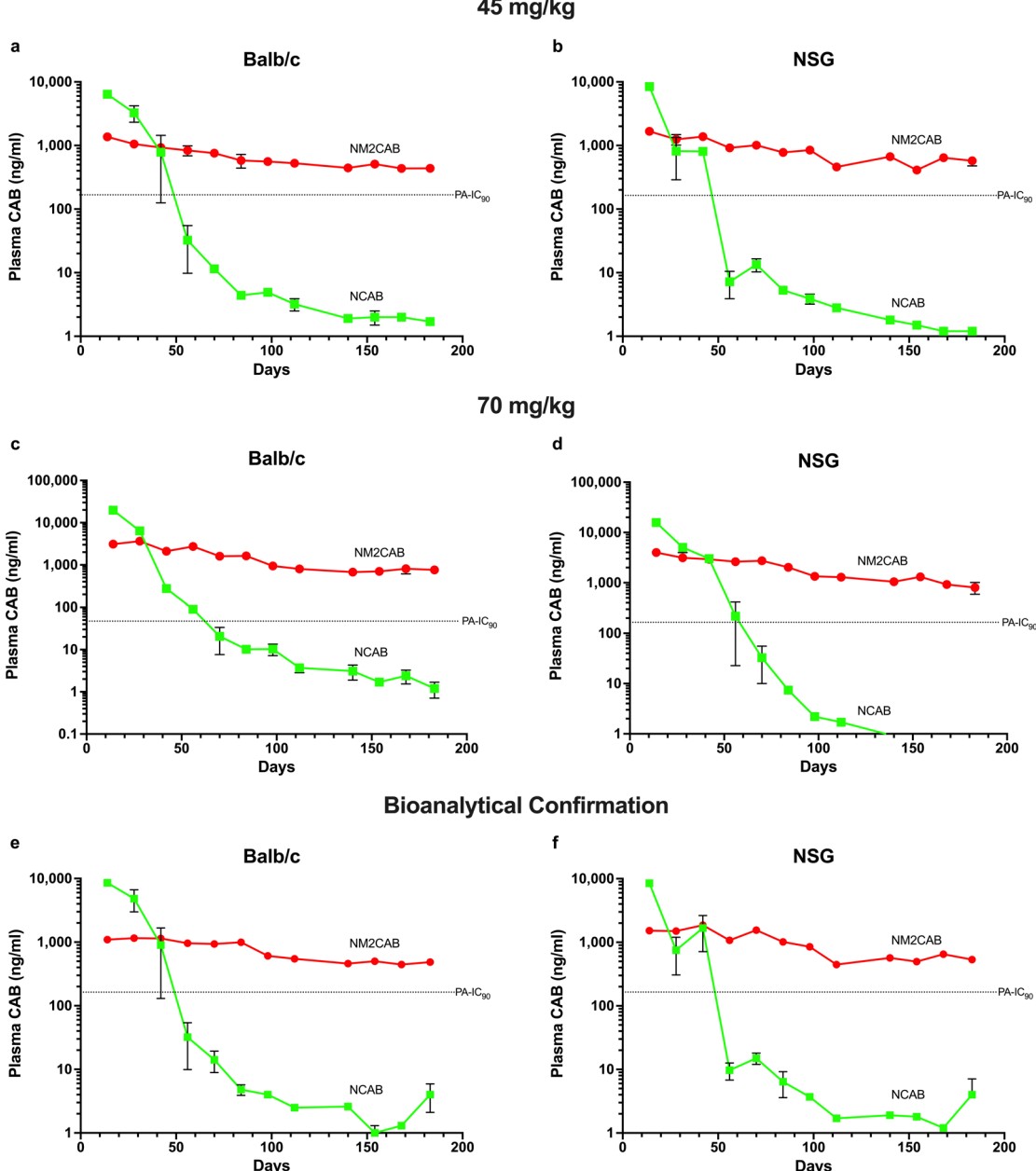

**Fig. 2 CAB drug plasma concentrations in Balb/c and NSG mice.** Male Balb/c or female NSG mice were given a single intramuscular injection of 45 or 70 mg CAB equivalents/kg of NCAB or NM2CAB. Plasma was collected every other week for 6 months at which time the study was terminated. CAB concentrations were quantitated by LC-MS/MS by Covance laboratories (**a**–**d**); treatment with 45 mg/kg in male Balb/c (**a**) and female NSG mice (**b**); treatment with 70 mg/kg in male Balb/c (**c**) and female NSG mice (**d**). CAB concentrations in plasma samples were also quantitated at UNMC (bioanalytical confirmation). Plasma CAB concentrations for the 45 mg/kg treatment groups are shown for male Balb/c (**e**) and female NSG mice (**f**). The dotted line indicates the CAB protein-adjusted $IC_{90}$ (PA-$IC_{90}$). Data are expressed as mean ± SEM ($N$ varied from 2 to 6 animals per group per time point due to loss of animals by natural causes, in both NCAB and NM2CAB treatment groups, or inadequate plasma volume collected during the study period, with exact values for each time point provided in the source data [10.6084/m9.figshare.14498025]).

higher dose (70 mg CAB equivalents/kg) provided a 2.2- to 2.8-fold increase in plasma CAB AUC compared to the lower dose in male Balb/cJ and female NSG mice, respectively (Fig. 2c, d). In addition, samples from the study were analyzed at UNMC using a different bioanalytical method and provided values that were similar to those obtained by the Covance analyses (Fig. 2e, f; Supplementary Fig. 2).

**Storage stability and release kinetics of NM2CAB and NCAB.** Storage stability was assessed by measuring total drug content,

while release kinetics were assessed by measuring released drug over time at room temperature. Both undiluted/original and 20×-diluted nanoformulations demonstrated storage stability and equivalent release kinetics at room temperature. The nanoformulations were used in the undiluted form for the animal studies while dilutions are made for all in vitro studies. NM2CAB nanoparticles were ~100% stable and demonstrated minimal drug release from the intact nanoparticles during storage for one year. We detected an up to 8% release of M2CAB from NM2CAB at day 0, without subsequent drug release for one year. During the

**Table 2 Storage stability of NCAB and NM2CAB nanoformulations.**

| | | NCAB | | NM2CAB | |
|---|---|---|---|---|---|
| | Time (Day) | Stability | % Release | Stability | % Release |
| Neat | 0 | 90.0% | 35.8% | 112.3% | 7.4% |
| | 4 | 88.9% | 38.4% | 95.2% | 6.7% |
| | 7 | 103.2% | 30.0% | 83.6% | 10.8% |
| | 14 | 102.4% | 26.0% | 92.7% | 7.0% |
| | 30 | 100.5% | 24.1% | 95.8% | 8.1% |
| | 60 | 95.8% | 39.3% | 108.4% | 6.2% |
| | 90 | 95.3% | 32.5% | 94.3% | 8.4% |
| | 252 | 95.9% | 37.6% | 112.2% | 6.3% |
| | 365 | 111.7% | 26.4% | 100.8% | 5.0% |
| 20×-diluted | 0 | 110.4% | 29.8% | 94.8% | 7.9% |
| | 4 | 106.5% | 26.5% | 84.3% | 9.7% |
| | 7 | 101.5% | 30.8% | 87.1% | 8.6% |
| | 14 | 111.3% | 23.7% | 106.8% | 6.1% |
| | 30 | 101.4% | 25.0% | 86.9% | 8.6% |
| | 60 | 114.8% | 20.3% | 116.1% | 7.0% |
| | 90 | 105.5% | 22.1% | 105.6% | 7.4% |
| | 252 | 102.7% | 31.1% | 102.3% | 8.5% |
| | 365 | 101.6% | 36.2% | 102.3% | 5.4% |

Stability of NCAB and NM2CAB at 36.7 and 55.4 mg/ml (undiluted), and 1.835 and 2.77 mg/ml (20×-diluted), respectively.

observation period, NM2CAB was completely stable (Table 2). In contrast, under the same storage conditions, a burst release of 35% was recorded for NCAB at day 0 without additional release for one year. Equivalent stability and release kinetics were seen in both 20×-diluted and undiluted formulations (Table 2).

**Chemical stability of unformulated M2CAB**. Next, we assessed the chemical stability of the solubilized form of M2CAB at a range of pHs [pH 1.0 (0.1 M HCl), pH 7.4 (PBS), pH 8.3 (heat-inactivated plasma) and pH 11.0 (0.1 M NaOH)]. In contrast to NM2CAB formulation, unformulated M2CAB was unstable at 37 °C over 24 h. At 24 h, percent M2CAB recovered was 87, 52, 5, and 2 at pH 1.0, pH 7.4, pH 8.3 and pH 11.0, respectively (Supplementary Fig. 3). Reductions in M2CAB prodrug concentrations paralleled native CAB formation, indicating the hydrolysis of the prodrug M2CAB into parent CAB.

**Prodrug and nanoformulation metabolic stability**. Native M2CAB, CAB, NM2CAB, and NCAB nanocrystals were evaluated for stability in mouse, rat, rabbit, monkey, dog, and human blood over 24 h (Fig. 3; Supplementary Fig. 4). The solubilized prodrug (M2CAB solution) was unstable in all tested species. Greater than 80% of M2CAB was hydrolyzed after 6 h in dog, monkey, mouse, rat, rabbit, and human (Fig. 3a, c, e). M2CAB was unstable in heat-inactivated plasma (Fig. 3g), indicating that its degradation in blood was due to chemical instability independent of a biological matrix. The decline in M2CAB prodrug paralleled the formation of native CAB over time (Fig. 3b, d, f, h). In contrast, NM2CAB was stable in blood from all species with >80% of M2CAB recovered and <10% of CAB formed after 24 h (Fig. 3a, c, e). Both CAB and NCAB were stable for 24 h in blood in all species (Supplementary Fig. 4). The metabolic stability of M2CAB solution was next assessed in liver S9 fractions from mouse, rat, rabbit, monkey, dog, and humans (Supplementary Fig. 5). In contrast to blood, <25% of M2CAB was lost in a 2 h incubation and was accounted for by CAB formation. No species-specific differences were observed. Similar to blood, chemical,

rather than biological factors, accounted for the instability. These reactions were next assessed in rat liver, spleen, muscle, lymph node, and heat-inactivated liver homogenate (Supplementary Fig. 6). In rat tissues, 50–80% of M2CAB solution remained after 6 h of incubation. M2CAB solution was also unstable in the control heat-inactivated liver homogenate, indicating that its degradation is due to chemical instability upon solubilization, and is independent of the biological matrices under similar assay conditions. The decline in M2CAB prodrug was accounted for by the formation of CAB in all tissues (Supplementary Fig. 6). In contrast to solubilized M2CAB, the prodrug within NM2CAB solid drug nanoparticles was stable in all tissues with 88% of M2CAB recovered and 10% of CAB formed after a 6 h incubation (Supplementary Fig. 6).

**Cellular uptake, retention, and prodrug release kinetics**. With the stability of the prodrug nanoformulation affirmed we next evaluated the efficiency of the prodrug nanoformulation for uptake, retention, and release rates in macrophages. Macrophages represent the primary cell type known to store the particles and the efficiency of uptake reflects the established depot of drug during the extended times observed in vivo[15]. To these ends, human monocyte-derived macrophages (MDMs) were used to record cell uptake, retention, and release of native-CAB, M2CAB, NCAB, and NM2CAB (Fig. 4). Up to 60, 30, 6, and 0.1% of NM2CAB, M2CAB, NCAB, CAB were taken by MDMs at 8 h, respectively. After cells were loaded for 8 h with each of the formulations they were evaluated for up to 30 days to determine drug retention and release. Seventy-five percent and 4% of intracellular NM2CAB and M2CAB were retained by MDMs at one month, respectively (Fig. 4a, c). Notably, intracellular CAB levels for NCAB and native CAB were below or at the limit of detection within hours of drug loading (Fig. 4a). M2CAB and CAB release from cells was determined by their appearance in culture media (Fig. 4b, d). Ninety-nine percent of the retained intracellular NM2CAB was in the form of the M2CAB prodrug (Fig. 4c), while 80% of the released extracellular NM2CAB was in the form of the parent CAB (Fig. 4b). We also tested stability and CAB formation for both NM2CAB and M2CAB in the macrophage cell culture media and in MDM-cell lysate (as a surrogate for intracellular stability) over 15 days. NM2CAB nanoparticles were <80% stable in media and 75% stable in cell lysates through 15 days, while M2CAB solution was completely hydrolyzed into CAB within 3 and 7 days in media and cell lysates, respectively. The data sets demonstrate the importance of assaying primary macrophages as reservoirs for prodrug nanoformulations highlighting the retention and release of both drug (CAB) and prodrug (M2CAB) in these cell sites.

**Blood drug measures**. Due to the instability of the solubilized form of M2CAB in blood, conditions were optimized to prevent ex vivo prodrug hydrolysis and CAB formation after sample collection. Methanol (containing 0.1% FA and 2.5 nM AF) inhibited the hydrolysis of the prodrug in blood. Therefore, we collected blood directly into tubes containing 0.1% FA and 2.5 nM AF in methanol to prevent any ex vivo M2CAB hydrolysis into CAB. A portion of the collected blood samples was saved to produce plasma. CAB blood concentrations were nearly 2-fold lower than that in plasma in all samples, and this was also shown previously[16,17] (Supplementary Fig. 1). Because blood concentrations of the prodrug were <1% of CAB, plasma samples seemed to be stabilized by EDTA, and because of the low-temperature storage (−20 °C), M2CAB blood and plasma concentrations were similar. Therefore, both plasma and blood sample collections are valid for M2CAB analysis and there was no

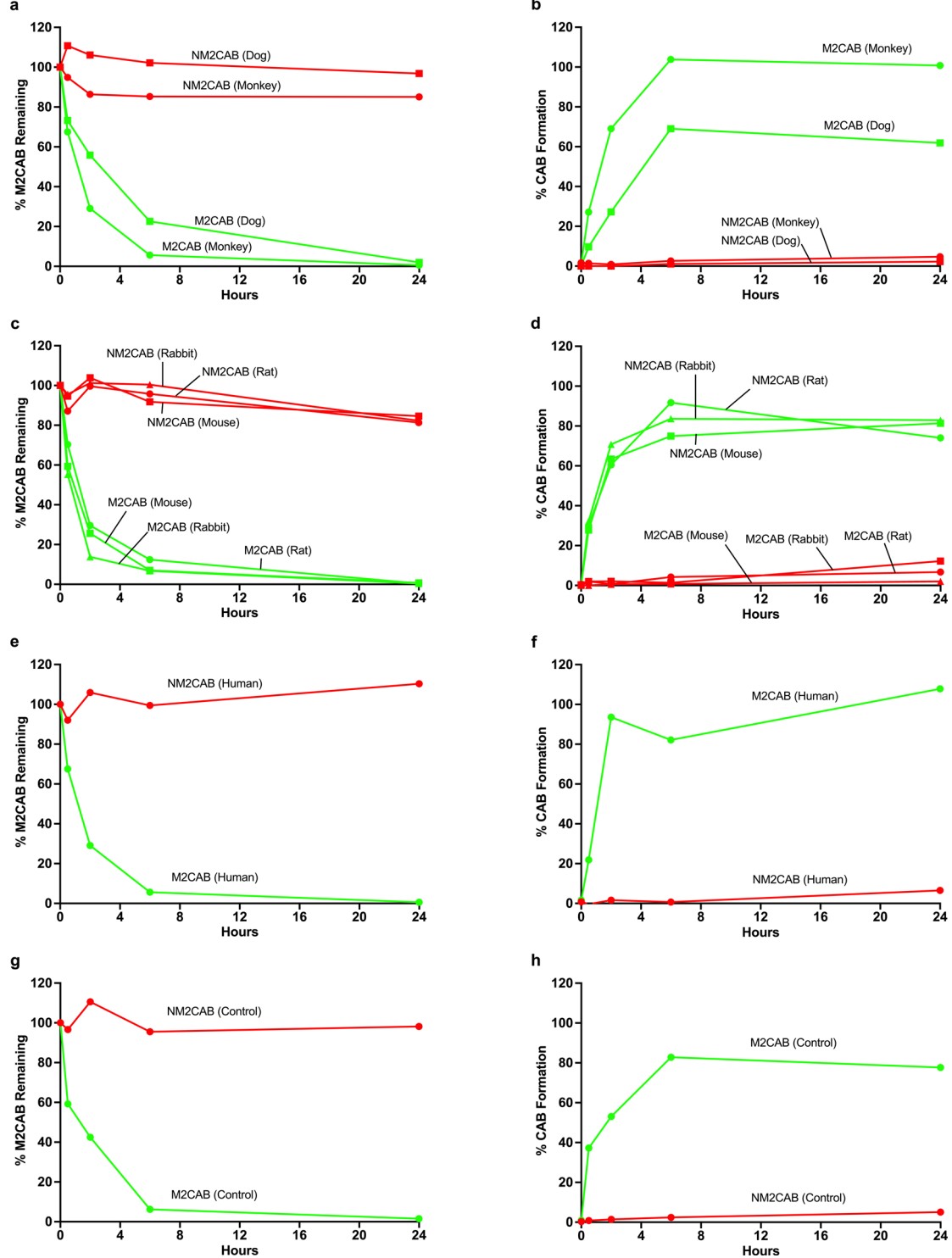

**Fig. 3 Blood metabolic stability profiles of prodrug (M2CAB) and nanoformulated prodrug (NM2CAB).** Stability profiles of prodrug (M2CAB) and nanoformulated prodrug (NM2CAB) at 1 μM concentration and formed native drug concentrations were determined in blood from six mammalian species (mouse, rat, rabbit, monkey, dog, and human) and in heat-inactivated plasma as a control. Data shown as disappearance for M2CAB over time (alone and from NM2CAB nanoformulations) in (**a**) monkey and dog, (**c**) mouse, rat and rabbit, (**e**) human, and (**g**) heat-inactivated plasma (control). Parallel measurements of CAB appearance over time were also recorded from M2CAB and NM2CAB in (**b**) monkey and dog, (**d**) mouse, rat and rabbit, (**f**) human, and (**h**) heat-inactivated plasma. Data are expressed as mean (*N* = 2 biological replicated per time point). Each experiment was repeated independently two times with equivalent results.

ex vivo hydrolysis of prodrug or parent drug formation during sample preparation, storage, and analysis.

**Tissue CAB distribution.** After both NM2CAB and NCAB treatment, CAB was widely distributed to all tissues with the

exception of the brain in both mice and rats. The highest CAB and M2CAB concentrations were at the site of injection and in lymph nodes, followed by the spleen, lung, kidney, and liver (Table 3). M2CAB concentrations in many tissues were higher than that of CAB at all time points. At day 28, CAB tissue

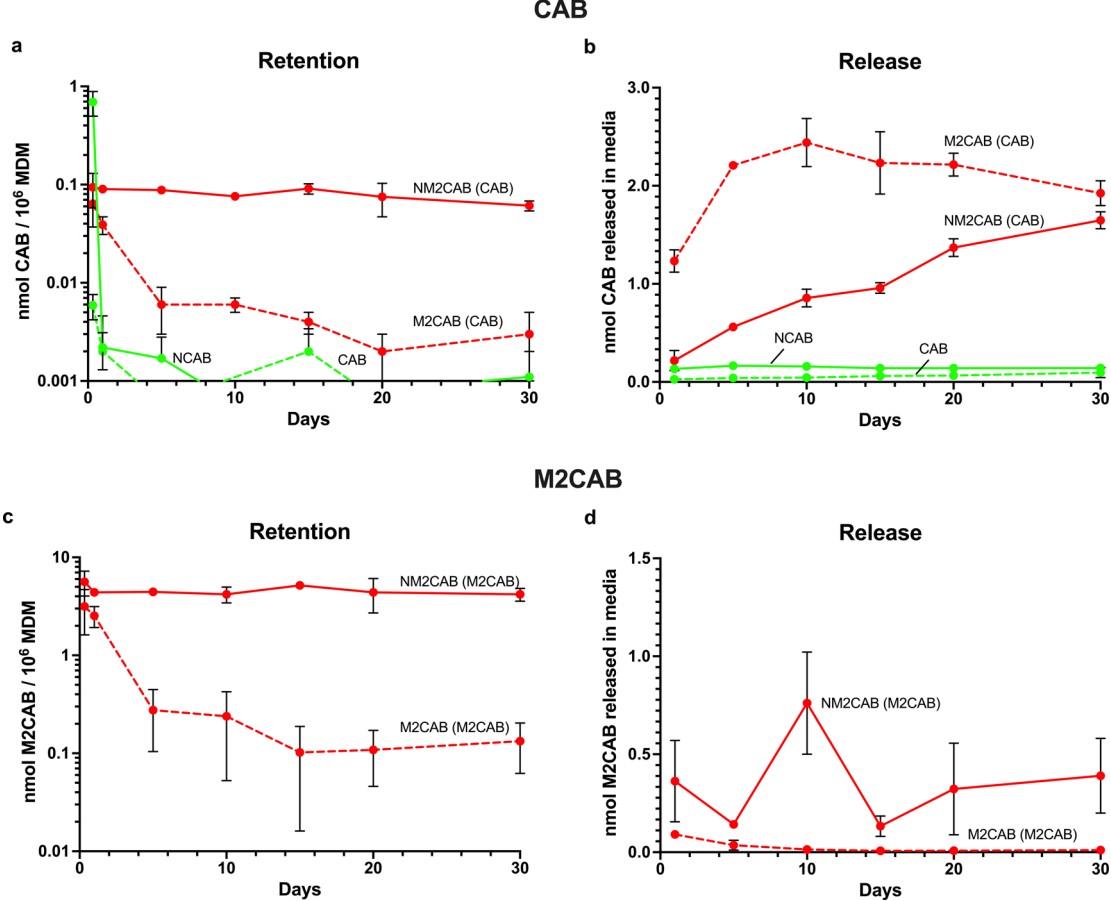

**Fig. 4 Cellular uptake, retention, and release of native drug and prodrug. a** Intracellular CAB, **b** CAB released in media; **c** intracellular M2CAB, and **d** M2CAB released in media after native-CAB, native-M2CAB, NCAB, and NM2CAB treatment to human monocyte-derived macrophages (MDM) at 10 μM (10 nmol/10⁶ cells) for 8 h. Data are expressed as mean ± SEM ($N = 3$ cell culture wells per time point). Each experiment was repeated independently a minimum of three times with similar results.

concentrations in mice were ~16,000, 3000, and 2200 ng/g in the site of injection, lymph nodes, and spleen, respectively, while CAB concentrations were in the range of 100–250 ng/g in other tissues (liver, kidney, lung, muscle, gut) and 16 ng/g in the brain. Similarly, M2CAB concentrations at day 28 were $1.7 \times 10^6$, 230,000, 35,000, and 3000 ng/g at the site of injection, lymph nodes, spleen, and liver, respectively, while in all other tissues it was in the range of 10–100 ng/g. By day 364 of NM2CAB administration in mice, CAB concentrations were 4939 and 86 ng/g at the site of injection and lymph nodes, respectively, while M2CAB concentrations in the same tissues were 419,000 and 4187 ng/g. The same trend in tissue distribution was observed after NCAB administration in mice, but with much lower CAB concentrations. By day 364, CAB concentration for NCAB at the site of injection was ~30 ng/g and undetected in all other tissues. Similarly, in rats the NM2CAB highest distribution was at the site of injection and lymph nodes. At day 28, CAB tissue concentrations were ~7000, 2200, and 1800 ng/g in the site of injection, lymph nodes, and lung, respectively, while CAB concentrations were in the range of 400-800 ng/g in other tissues (liver, spleen, kidney, muscle, gut) and 97 ng/g in the brain. Similarly, M2CAB concentrations at day 28 were $7.6 \times 10^6$, 458, and 339 ng/g at the site of injection, liver, and spleen, respectively, while in all other tissues were <40 ng/g. By day 364 after NM2CAB administration in rats, CAB concentrations were 1800 and 118 ng/g in the site of injection and lymph nodes, respectively, while M2CAB concentrations in the same tissues were 103,000 and 1.2 ng/g. The brain showed the lowest tissue

distribution in both species after both NM2CAB and NCAB administration. By day 168, CAB brain concentrations were barely detectable after NCAB administration in both mice and rats, while NM2CAB administration produced ~8 and 44 ng/g in mice and rats, respectively. Also, the prodrug (M2CAB) brain concentrations were below the detection limit in both mice and rats at all time points. CAB tissue concentrations were not only higher after NM2CAB compared to NCAB, but they also decreased at a much slower rate over time. Day 168 concentrations were 50 and <1% on day 28 after NM2CAB and NCAB administration, respectively (Table 3).

## Discussion

CAB is a second-generation INSTI currently nearing USA FDA approval as a LA injectable[18]. While effective in PrEP and has generated attention by the scientific and patient community for both treatment and prevention of HIV/AIDS[12,17,19] limitations in dosing volumes, intervals of administration, and local injection site reactions remain. To overcome these, prodrug nanoformulated were developed and are currently undergoing pre-clinical evaluations by our group[14,15]. Early proof of concept studies showed encouraging results. However, what was done previously in support of NM2CAB required affirmation and safety evaluation to better position the product for future clinical testing. Underlying mechanisms linked to a year-long medicine also required evaluation. Towards both ends, independent laboratories employed two strains of mice and rats to affirm our prior works.

**Table 3 CAB and M2CAB tissue distribution.**

| Animal | Time (Day) | Liver Mean (ng/g) | Liver SEM | Spleen Mean (ng/g) | Spleen SEM | Kidney Mean (ng/g) | Kidney SEM | Brain Mean (ng/g) | Brain SEM | Lung Mean (ng/g) | Lung SEM | Muscle Mean (ng/g) | Muscle SEM | Gut Mean (ng/g) | Gut SEM | Lymph Node Mean (ng/g) | Lymph Node SEM | Site of Injection Mean (ng/g) | Site of Injection SEM |
|---|---|---|---|---|---|---|---|---|---|---|---|---|---|---|---|---|---|---|---|
| Mouse | CAB concentrations from NCAB | | | | | | | | | | | | | | | | | | |
| | Day 28 | 76.7 | 10.6 | 253.3 | 58.1 | 217.2 | 49.6 | 14.6 | 3.9 | 753.8 | 257.5 | 123.3 | 30.0 | 149.4 | 34.7 | 621 | 150 | 16,437 | 7127 |
| | Day 84 | 0.6 | 0.3 | 1.0 | 0.2 | 2.1 | 0.7 | 0.0 | 0.0 | 3.0 | 0.2 | 0.5 | 0.1 | 1.7 | 0.9 | 15 | 5 | 306 | 50 |
| | Day 168 | 0.0 | 0.0 | 0.1 | 0.1 | 0.4 | 0.1 | 0.0 | 0.0 | 4.8 | 0.9 | 0.6 | 0.3 | 0.5 | 0.3 | 5 | 2 | 169 | 11 |
| | Day 364 | 0.0 | 0.0 | 0.0 | 0.0 | 0.4 | 0.3 | 0.0 | 0.0 | 0.0 | 0.0 | 0.4 | 0.3 | 0.0 | 0.0 | 0 | 0 | 30 | 4 |
| | CAB concentrations from NM2CAB | | | | | | | | | | | | | | | | | | |
| | Day 28 | 159.3 | 30.7 | 2201.3 | 202.4 | 196.6 | 15.0 | 16.4 | 2.4 | 224.9 | 21.4 | 103.9 | 12.7 | 158.2 | 19.4 | 2,995 | 964 | 16,558 | 2490 |
| | Day 84 | 76.3 | 12.5 | 1268.5 | 466.8 | 133.3 | 17.4 | 10.5 | 1.6 | 254.9 | 82.2 | 56.8 | 7.5 | 88.0 | 6.2 | 3,394 | 871 | 9946 | 1675 |
| | Day 168 | 44.9 | 2.0 | 59.7 | 6.8 | 148.4 | 61.5 | 8.1 | 1.3 | 132.5 | 14.9 | 36.0 | 4.8 | 56.7 | 3.6 | 1,580 | 568 | 9749 | 621 |
| | Day 364 | 14.9 | 0.5 | 16.4 | 1.2 | 36.1 | 3.5 | 0.5 | 0.2 | 48.0 | 3.8 | 11.0 | 1.7 | 20.4 | 2.7 | 86 | 20 | 4939 | 708 |
| | M2CAB concentrations from NM2CAB | | | | | | | | | | | | | | | | | | |
| | Day 28 | 3208.7 | 827.1 | 35,933 | 8914 | 11.4 | 3.4 | 4.5 | 1.8 | 137.1 | 38.5 | 8.9 | 4.4 | 9.4 | 2.6 | 231,685 | 135,071 | 1,757,500 | 259,624 |
| | Day 84 | 558.6 | 271.4 | 13,438 | 4543 | 1.7 | 0.5 | 0.5 | 0.1 | 53.7 | 26.4 | 8.8 | 3.8 | 3.7 | 1.5 | 274,090 | 98,711 | 741,500 | 175,267 |
| | Day 168 | 39.8 | 6.9 | 492 | 50 | 1.1 | 0.3 | 5.8 | 1.9 | 12.7 | 3.8 | 1.2 | 0.4 | 2.4 | 0.8 | 81,752 | 33,916 | 756,333 | 83,539 |
| | Day 364 | 5.3 | 0.7 | 33 | 4 | 2.2 | 1.4 | 0.0 | 0.0 | 3.5 | 2.3 | 1.8 | 1.4 | 3.3 | 1.9 | 4,188 | 1,550 | 419,000 | 36,491 |
| Rat | CAB concentrations from NCAB | | | | | | | | | | | | | | | | | | |
| | Day 28 | 1411.2 | 296.5 | 855.1 | 206.0 | 2351.6 | 624.3 | 216.0 | 54.1 | 2747.8 | 667.9 | 3,071.2 | 443.3 | 1,086.0 | 262.5 | 3,105 | 488 | 675,486 | 250,918 |
| | Day 84 | 55.6 | 15.4 | 124.2 | 39.4 | 164.4 | 35.1 | 13.3 | 4.5 | 420.4 | 203.0 | 300.9 | 74.6 | 127.4 | 44.1 | 485 | 182 | 1219 | 956 |
| | Day 168 | 0.6 | 0.2 | 3.0 | 1.0 | 39.8 | 5.1 | 1.6 | 0.9 | 3.7 | 0.8 | 6.6 | 1.9 | 9.6 | 3.3 | 7 | 1 | 43 | 28 |
| | Day 364 | 0.0 | 0.0 | 0.0 | 0.0 | 0.5 | 0.2 | 0.0 | 0.0 | 2.5 | 1.1 | 0.0 | 0.0 | 0.1 | 0.0 | 0 | 0 | 16 | 1 |
| | CAB concentrations from NM2CAB | | | | | | | | | | | | | | | | | | |
| | Day 28 | 674.1 | 184.6 | 434.1 | 84.3 | 779.0 | 158.1 | 97.7 | 19.2 | 1824.5 | 375.4 | 560.5 | 110.5 | 692.7 | 126.8 | 2,210 | 401 | 6821 | 693 |
| | Day 84 | 331.0 | 61.2 | 473.2 | 68.9 | 552.7 | 78.7 | 74.3 | 15.3 | 1490.7 | 253.5 | 506.6 | 112.1 | 636.8 | 128.9 | 1,411 | 224 | 4893 | 990 |
| | Day 168 | 177.0 | 53.8 | 126.3 | 27.7 | 323.9 | 70.5 | 43.7 | 10.0 | 334.1 | 52.1 | 228.6 | 67.2 | 175.7 | 31.2 | 723 | 171 | 1915 | 449 |
| | Day 364 | 44.4 | 14.0 | 44.2 | 16.4 | 152.0 | 41.4 | 5.4 | 2.6 | 149.4 | 61.8 | 26.4 | 9.9 | 54.5 | 18.1 | 119 | 49 | 1795 | 261 |
| | M2CAB concentrations from NM2CAB | | | | | | | | | | | | | | | | | | |
| | Day 28 | 458.1 | 66.9 | 339.3 | 48.3 | 1.9 | 0.2 | 0.5 | 0.1 | 37.5 | 28.3 | 27.8 | 18.1 | 35.8 | 10.5 | 12 | 3 | 762,500 | 72,235 |
| | Day 84 | 264.4 | 90.1 | 302.0 | 58.4 | 0.7 | 0.1 | 0.3 | 0.1 | 24.6 | 4.9 | 0.8 | 0.2 | 4.3 | 0.6 | 9 | 1 | 240,566 | 89,748 |
| | Day 168 | 51.9 | 9.9 | 89.2 | 22.5 | 1.3 | 0.2 | 0.6 | 0.1 | 5.9 | 1.2 | 0.6 | 0.2 | 2.5 | 0.5 | 8 | 2 | 108,000 | 40,875 |
| | Day 364 | 7.9 | 2.4 | 8.8 | 2.8 | 2.7 | 2.7 | 0.0 | 0.0 | 0.4 | 0.4 | 0.6 | 0.3 | 0.0 | 0.0 | 1 | 0 | 102,937 | 15,663 |

CAB (native) and M2CAB (prodrug) tissue distribution in Balb/cJ mice and SD rats after a single IM injection of 45 mg CAB equivalents/kg NCAB and NM2CAB ($N = 6$; mean ± SEM).

Results showed sustained CAB plasma levels for at least one year in all species. Plasma concentrations were up to 100-fold higher than those resulting from NCAB (an equivalent preparation to the CAB LA). Most notably we now show that the mechanism of this extended half-life was attributed to prodrug release from the solid nanocrystals stored in tissues rather than the stability (chemical and enzymatic) of the prodrug itself. These findings offer insights into how best to bring such extended-release medicines to the clinic with broad relevance to ARVs now being modified as LA formulations.

INSTIs are a newer class of antiretrovirals (ARVs) that prevent the integration of viral DNA into the host cell DNA, by inhibiting the viral integrase enzyme[20]. The main advantage of INSTIs rests in their activity against HIV mutant viruses resistant to other ARV classes[21]. Raltegravir (RAL) was the first INSTI approved by the USA FDA in 2007; elvitegravir (EVG), dolutegravir (DTG), and bictegravir (BIC), were next. CAB is the newest addition to this class, and is currently under development by ViiV Healthcare, Brentford, Middlesex UK[18].

CAB has more favorable PK properties than other INSTIs. This includes its long plasma half-life of 30–40 h[8], fast oral absorption ($t_{max} = 1.5$–2 h)[16], near-complete oral bioavailability[16,22], and low DDI potential due to its high membrane permeability and low affinity for CYP450[9,10]. In humans, CAB is primarily eliminated unchanged in feces (up to 50% unchanged and 10% unidentified metabolites) and as glucuronide metabolites in urine (27%)[16]. In addition, CAB is characterized by low aqueous solubility (0.015 mg/mL in water), and high melting point (248–251 °C), which permits its loading at high concentrations as nanosuspensions[11]. CAB LA is in late-stage clinical trials[12]. When administered as a LA parenteral, CAB concentrations remain above four times the PA-IC$_{90}$ of 0.664 ug/ml, for 16 weeks after an 800 mg loading dose[13]. With a drug load of 400 mg CAB can maintain plasma drug levels beyond four times the PA-IC$_{90}$ at 8 weeks[23]. The combination of CAB and rilpivirine LA in infected adults (the LATTE-2, Long-Acting antireTroviral Treatment Enabling trial) showed that 400, 600, and 800 mg administered every 2, 8, and 12 weeks was as effective as once daily 30 mg CAB given orally. Therefore, formulations that allow longer dosing intervals and/or lower dosing volumes can markedly improve what is now available for CAB LA.

To this end, our group has recently developed several lipophilic fatty acid ester-CAB prodrugs (MCAB) prepared as LA nanoformulations. NMCAB and NM2CAB markedly improved and extended the PK profile of CAB compared to CAB LA[15]. In this paper, we expanded on our earlier work by reporting the PK and BD of the lead nanoformulated CAB prodrug (NM2CAB) in mice and rats and compared it to NCAB, a formulation reflective of CAB LA. Our year-long NM2CAB was reproducible and robust as demonstrated by results from three independent laboratories in two strains of mice and in three animal species including mice, rats, and rhesus macaques. Also, we elucidated the mechanisms behind the enhanced PK profiles of NM2CAB using in vitro systems for absorption, distribution, metabolism, and excretion and release kinetics.

All together, we characterized the metabolic and chemical stability, release kinetics, as well as PK profile of NM2CAB in comparison to NCAB. We used several means to investigate the mechanisms behind the enhanced PK profile of NM2CAB. First, the chemical stability and release kinetics of NM2CAB were assessed over one year. The original and diluted NM2CAB were stable with a minimal burst release of 8% at day 0 without any further release. Second, we showed that NCAB showed similar results, but with a higher burst release of 35%. Third, the stability of M2CAB in solubilized form was pH-dependent, where degradation to CAB accelerated with increasing pH and was minimal at pH 1.0. The NM2CAB formulations produced were stable for 360 days with respect to size (nm), polydispersity (PDI) and surface charge (zeta potential) (Supplementary Fig. 7).

M2CAB is an ester-linked prodrug, which is hydrolyzed in tissues and blood[15,24], presumably by esterases and pH-dependent chemical hydrolysis upon release and solubilization of the lipophilic prodrug. Species differences in the distribution of these esterases have been previously reported[24]. Therefore, we quantified the metabolic stability of CAB, M2CAB, NCAB, and NM2CAB in blood and liver S9 fractions from five animal species and humans. In blood, the ester bond of the M2CAB prodrug was rapidly hydrolyzed into the parent CAB within 6 h in all species. In addition, M2CAB solution was unstable in the control heat-inactivated plasma and in PBS, indicating chemical instability, as well. In contrast, NM2CAB solid drug nanocrystals, CAB, and NCAB were stable in blood and more than 90% of M2CAB (NM2CAB) or CAB (CAB, NCAB) was recovered by 24 h in all six species. M2CAB solution was also hydrolyzed in liver S9 fraction from all species, but at a lower rate than in the blood. In addition, the metabolic stability of NM2CAB was quantified in several tissue (liver, spleen, muscle, lymph node) homogenates from rats. In general, the hydrolysis rate of M2CAB was slower in tissues than in blood.

Our mechanistic studies were also performed in MDMs. Macrophages serve as a major depot for CAB prodrug nanoformulations while at the same time a principal HIV-1 reservoir[25–27]. Therefore, we quantified the uptake, retention, and release kinetics of NM2CAB in MDMs. NM2CAB was efficiently taken up and retained for at least 30 days by MDMs. NM2CAB demonstrated up to 100-fold greater uptake and retention by MDMs compared to NCAB and unformulated CAB. While unformulated solution of M2CAB prodrug demonstrated enhanced cellular uptake, it was not retained inside the cells and <95% was released into the media within days and was recovered as CAB. M2CAB solution hydrolyzed into CAB both intracellularly and in the media while NM2CAB nanoparticles were stable both intracellularly and in the media, and slowly released M2CAB over 30 days. Collectively, these data show that NM2CAB solid drug nanocrystals enhance cellular drug uptake and extend intracellular drug retention due to the slow release and solubilization of M2CAB.

PK studies were performed in mice, rats, and rhesus macaques. A single dose of NM2CAB demonstrated substantial improvements in CAB PK profile, reflected by elevated and sustained plasma and tissue CAB concentrations compared to NCAB. CAB terminal $t_{0.5}$ was extended from 7.5 to 118.4 days in mice and from 15.7 to 81.5 days in rats after NM2CAB administration compared to NCAB. CAB plasma levels were maintained above the PA-IC$_{90}$ of 166 ng/ml for at least one year after a single dose of NM2CAB to both mice and rats. By the end of the study, CAB plasma levels were 200- to 300-fold higher after NM2CAB compared to N2CAB in rodents. Similarly, NM2CAB was slowly eliminated ($t_{0.5} = 184$ days) from plasma. CAB was detected in plasma for two years in monkeys.

In all species tested, NM2CAB and NCAB were widely distributed to all tissues with the exception of the brain. The highest concentrations of both CAB and M2CAB were detected at the site of injection in muscle as well as in lymph nodes and spleen. After NM2CAB administration in mice, M2CAB concentrations were 10- to 100-times higher compared to CAB in lymph nodes and spleen and 50- to 100-times higher at the site of injection. Similar to plasma, CAB tissue levels were significantly higher after NM2CAB administration compared to NCAB in both mice and rats. Furthermore, the difference in tissue levels increased over time. By the end of the study, CAB levels in all tested tissues after NM2CAB treatment were up to 4000-times higher compared to those after NCAB.

Hydrolysis of the prodrug M2CAB, after release from NM2CAB, located in plasma and/or and tissues can be the source for the parent CAB detected in plasma. However, plasma prodrug levels after NM2CAB injection were low, while tissue, including the site of injection, exhibited high prodrug and parent drug concentrations for the entire study period. In contrast, CAB tissue concentrations after NCAB administration rapidly declined to undetectable levels within five weeks. Therefore, the source of the high and sustained plasma CAB concentrations associated with NM2CAB administration is dependent on the rate of nanocrystal dissolution and prodrug release in tissue and at the injection sites and the hydrolysis rate of the ester chemical bond proved rapid after the surfactant coated nanocrystals were transformed into solution. Thus, it can be concluded that the sustained blood and tissue PK in vivo is due to the stability of NM2CAB and slow M2CAB dissolution at tissue and injection sites over time periods up to a year.

To confirm our PK findings, we repeated the male Balb/cJ mouse study in females of another mouse strain (NSG). In addition, this study was performed by three independent laboratories (the UNMC College of Pharmacy, the UNMC College of Medicine[15], and Covance Laboratories). Finally, samples generated from the Covance Balb/cJ mouse study were quantified by Covance and by our laboratory using completely different LC-MS/MS methods. Results from all these studies were in agreement. CAB plasma concentrations were initially higher up to 6 weeks after NCAB administration followed by a faster decline compared to NM2CAB. CAB was detected for the entire period of the study after NM2CAB administration, while it was generally undetected or around the detection limit by 3 months after M2CAB administration. At all time points after 6 weeks where CAB was detected, its concentration was 100- to 1000-fold higher after NM2CAB compared to NCAB administration. The $t_{0.5}$ of CAB was extended from 3–8 days to 65–131 days after NM2CAB administration compared to NCAB. Finally, reproducible results were obtained for the study samples analyzed by two independent laboratories using different LC-MS/MS methods. Collectively, this provides conclusive evidence that the enhanced PK properties of NM2CAB over at least one year is reproducible in both sexes and in two strains of mice (Balb/cJ and NSG) by three different laboratories one of which is a contract facility and independent of the bioanalysis performed by the two academic facilities.

A limitation of our NM2CAB NHP study rests in the screening of a single, low CAB dose. It is also worth noting that prior studies demonstrated that the terminal phase half-life of CAB in macaques is shorter (3–12 days) compared to humans (21–50 days)[28–30]. It has been shown in NHP that plasma CAB concentrations of >3×PA-IC$_{90}$ and ≥1×PA-IC$_{90}$ confer 100 and 97% protection against intrarectal SHIV challenge, respectively[31]. The rapid clearance rate of CAB in NHP required that a higher dose of 50 mg/kg be used in NHP to achieve 4×PA-IC$_{90}$ in plasma. Notably, using dose extrapolation, for a 70 kg person, a human equivalent dose of 50 mg/kg in NHP used for PrEP studies translates to 1129 mg of CAB LA in humans[32]. This is notably higher than the 400 and 600 mg that has now been extensively characterized in humans for monthly or bimonthly dosing and shown to be effective. These data underscore interspecies differences in CAB metabolism. For our study, a lower dose of 45 mg/kg was used and therefore dose-ranging studies will be evaluated in our future studies. Notably, our NM2CAB PK data in mice revealed a dose-dependent increase in CAB suggesting that dose adjustment will be needed for future NHP experiments. Dose escalation studies at lower dosing volumes will be evaluated taking into consideration a current formulation of 400 mg/ml drug concentration. To this end the following steps will be taken. First, we will extend analyses of the M2CAB

prodrug and its CAB metabolite in dose-escalating studies performed in vivo in mice, rats, and NHPs. Second, each of these preclinical evaluations will include tissue drug distribution and clearance so that CAB persistence in lymphoid tissues would serve as a gauge for clinical responses. Third, confirmatory testing need include, but not limited to, plasma protein binding and hepatic microsomal metabolic stability. The AUC$_{0-\infty}$ of NM2CAB at 5–500 mg/kg will determine PK profiles obtained at varying dose. Fourth, as the NM2CAB prodrug, can be quickly hydrolyzed into CAB and is protected by the nanofomulation after intramuscular administration the mechanisms uncovered in this current report will serve as the foundation for these future studies. Fifth, as NM2CAB easily readily penetrates lymphoid tissue the concentrations in gut, lymph nodes, spleen, and brain at each of these doses will need to be considered. The tested 45 mg/kg dose and completed in these studies translates to a 508 mg human equivalent CAB dose administered over a year. While we understand that NM2CAB is a promising candidate prodrug and appreciate its complexities these studies are slated to be completed as a research translational priority.

In summary, we demonstrate both rigor and reproducibility of our year-long CAB prodrug formulation. NM2CAB administration to two strains of mice, rats, and monkeys resulted in enhanced and sustained CAB plasma levels for at least one year. Plasma concentrations were up to, or greater than, 100-times those resulting from the equivalent NCAB (a CAB LA equivalent formulation). The advantages of NM2CAB for LA treatment are several-fold. NM2CAB can be manufactured at a drug concentration of 400 mg/ml, twice the CAB LA concentration of 200 mg/ml. Thus, NM2CAB would require a 1 ml injection volume for yearly dosing when compared to a 2 ml monthly injection for CAB LA. In addition, no injection site reactions were observed. Thus, NM2CAB provides an improvement over the current approved CAB LA with an extended dosing interval, reduced injection volume, and no injection site reactions. Importantly, unlike orally administered medicines where clearance rates determine the terminal drug half-life, the extended duration of NM2CAB is dependent on the amount of drug absorbed/day from tissue depot sites. These enable a lower administered dose to last for a year. We also elucidated the mechanisms behind this enhanced PK profile from tissue distribution and in vitro data. NM2CAB accumulates and is retained in the injection site and in tissues, which serve as depots that slowly release the prodrug over months to years. In vitro data suggest that the observed enhanced tissue accumulation of NM2CAB is due to enhanced cellular uptake of the nanocrystals from the injection site, intracellular retention, and nanocrystal stability in macrophages. In tissues, M2CAB undergoes slow dissociation from the nanoformulation which then is rapidly hydrolyzed into native CAB by both chemical and enzymatic hydrolysis. This was cross validated by chemical and metabolic studies in vitro using blood and tissue homogenates. Overall, the integration of mechanistic in vitro and in vivo data sets has elucidated the mechanisms behind the enhanced and sustained PK behavior of NM2CAB. These data sets to support the premise that enhanced macrophage nanocrystal delivery and formation of intracellular tissue drug depots and slow prodrug dissolution rate contribute to the extended NM2CAB apparent drug half-life.

## Methods

**Chemicals**. CAB was purchased from BOC Sciences (Shirley, NY, USA). Methanol (LC/MS Grade), acetonitrile (LC/MS Grade), water (LC/MS Grade), formic acid (FA, LC/MS Grade), ammonium formate (AF), bovine serum albumin (BSA), and phosphate-buffered saline (PBS) were obtained from Fisher Scientific (Fair Lawn, NJ, USA). Blood from mouse, rat, rabbit, monkey, dog, and human was purchased from Innovative Research (Novi, MI, USA). Dulbecco's Modification of Eagle's

Medium (DMEM) was purchased from Corning Life Sciences (Tewksbury, MA, USA).

**Nanoformulation preparation and characterization.** Both CAB (NCAB) and M2CAB prodrug (NM2CAB) nanoformulations were prepared by high-pressure homogenization in the Nebraska Nanomedicine Production Plant (NNPP) using good laboratory practices (GLP) protocols. Briefly, each solid drug/prodrug was dispersed in a poloxamer 407 (P407) solution at a drug/prodrug to surfactant ratio of 10:1 (w/w) in endotoxin-free water and allowed to form a presuspension. The presuspension was homogenized on an Avestin EmulsiFlex-C3 high-pressure homogenizer (Ottawa, ON, Canada) at $20,000 \pm 1000$ PSI until the desired particle size was achieved. The formulations were characterized for particle size, polydispersity index (PDI), and zeta potential by dynamic light scattering (DLS) using a Malvern Zetasizer Nano-ZSP (Worcestershire, UK). Drug loading was determined by ultra-performance liquid chromatography-tandem mass spectrometry (LC-MS/MS)[15]. Endotoxin concentrations were determined using a Charles River Endosafe nexgen-PTS system (Charles River, USA) and only formulations with endotoxin levels <5 EU/kg were used for animal studies.

**Storage stability and release kinetics.** Two concentrations were studied including the original formulation after manufacturing (undiluted; 55.4 mg/ml for NM2CAB and 36.7 mg/ml for NCAB) and 20-fold (20×) diluted formulation (2.77 mg/ml for NM2CAB and 1.83 mg/ml for NCAB) in water. The original and 20X dilutions represent what was used for the rodent and in vitro studies, respectively. Nanoformulations were stored at room temperature and samples were collected and both total and release drug concentrations were measured on days 0, 4, 7, 14, 30, 60, 90, 252, and 365.

For quantification of the released drug, aliquots were collected at the specified time points and diluted in 4% BSA solution to a final drug concentration of 1 µg/ml, then samples were centrifuged at $10,000 \times g$ for 10 min to pellet and separate intact nanoparticles from the released drug in the supernatant. Fifty microliters of aliquots were collected before (total drug concentration) and after (released drug concentration in the supernatant) centrifugation in 7 volumes of methanol (containing 0.1% FA and 2.5 mM AF). Release was calculated as the amount of drug in supernatant: the total amount of drug in the incubation at each time point. In addition, a similar control (mass balance) set was aliquoted in separate individual vials (2 vials per time point) at day 0 to allow the determination of total drug content at every time point. Stability was calculated as total amount of drug in the system at every time point to that at day 0. CAB from NCAB incubations and M2CAB from NM2CAB incubations were quantified by LC-MS/MS.

**Metabolic stability.** To determine differences in rate of CAB and M2CAB stability in blood across species, 100 µl mouse, rat, rabbit, dog, monkey, and human blood was incubated with 1 µM native-CAB, native-M2CAB, NCAB, or NM2CAB at 37 °C. At different time points (0, 30 min, 2 h and 6 h), 0.9 ml of methanol (containing 0.1% FA and 2.5 mM AF) was added to each sample and vortexed for 3 min. The samples were then centrifuged at $16,000 \times g$ for 10 min after which 10 µl of the supernatant was mixed with 80% methanol (containing 0.1% FA and 2.5 mM AF) containing an internal standard (IS) and analyzed by LC-MS/MS. For the time zero start time, a 100 µl of ice-cold blood was spiked with M2CAB and immediately 0.9 ml of methanol (containing 0.1% FA and 2.5 mM AF) was added. Heat-inactivated plasma was incubated at the same conditions and used as negative controls to differentiate chemical versus metabolic stability.

For metabolic stability in tissues, freshly collected rat liver, spleen, muscle, and lymph node tissues were homogenized in five-volumes of PBS (w/v). One hundred µl tissue homogenates were incubated with 1 µM M2CAB or NM2CAB at 37 °C. At different time points (0, 30 min, 2 h, and 6 h), 0.9 ml of methanol (containing 0.1% FA and 2.5 mM AF) was added to each sample and vortexed for 3 min. Samples were centrifuged at $16,000 \times g$ for 10 min, 10 µl supernatant was mixed with 80% methanol (containing 0.1% FA and 2.5 mM AF) containing IS and analyzed by LC-MS/MS.

The metabolic stabilities of CAB and M2CAB were determined using mouse, rat, rabbit, dog, monkey, and human liver S9 fractions (XenoTech, LLC, Lenexa, KS, USA). Briefly, CAB and M2CAB were incubated at 1 µM in a 100 µl mixture containing S9 fractions at 1 mg/ml protein concentration, NADPH (1 mM), saccharolactone (5 mM), uridine 5′-diphospho-glucuronic acid (UDPGA) (1 mM), and 3′-phosphoadenosin-5′-phosphosulphate (PAPS) (0.1 M) in PBS (100 mM, pH 7.4) at 37 °C. For each time point (0, 30, and 120 min), reactions were quenched by adding 0.9 ml of methanol (containing 0.1% FA and 2.5 mM AF), followed by centrifugation at $15,000 \times g$ for 10 min. Ten microliters of the supernatants were then analyzed by LC-MS/MS. Heat-inactivated S9 fractions were incubated at the same conditions and used as negative controls. Hydroxycoumarin (7-HC) metabolites including 7-HC-glucuronide and 7-HC-sulfate, and the testosterone metabolite 6β-hydroxytestosterone were used as positive controls for phase I and phase II metabolism.

**Cellular uptake, retention, and drug release.** Human peripheral blood monocytes were obtained by leukapheresis from HIV-1,2 and hepatitis B seronegative donors and purified by centrifugal elutriation. Monocytes were cultured for the first seven

days in macrophage colony stimulating factor enriched media (MCSF, 1000 U/ml) to facilitate cell vitality and differentiation into macrophages[15,29]. MDM were cultured in DMEM containing 4.5 g/L glucose, L-glutamine, and sodium pyruvate, and supplemented with 10% heat-inactivated human serum, 50 µg/ml gentamicin, and 10 µg/ml ciprofloxacin. Cells were maintained in clear flat-bottom 12-well plates at 37 °C in a 5% $CO_2$ incubator at a density of $1.0 \times 10^6$ cells per well. Half-culture media was replaced with fresh media every other day. After differentiation, MDM were utilized for CAB prodrug or drug nanoparticle uptake, retention and release. To quantify cellular uptake and retention kinetics, MDMs were incubated with 10 µM native-CAB, NCAB, native-M2CAB or NM2CAB. For uptake, MDM were collected at 8 h following treatment to measure intracellular drug and prodrug levels. For retention studies, after 8 h uptake, loaded MDMs were washed twice with PBS and fresh culture medium was added, and half-media was replaced every other day until the cells were harvested. At days 1, 5, 10, 15, 20, and 30, adherent MDM were washed twice with PBS, scraped into PBS, and counted using an Invitrogen Countess Automated Cell Counter (Carlsbad, CA). Cell suspension was then centrifuged at $900 \times g$ for 8 min at 4 °C. Cell pellets were sonicated in 200 µl of 80% methanol (containing 0.1% FA and 2.5 mM AF) to extract intracellular drug. The resultant lysates were centrifuged at $15,000 \times g$ for 10 min at 4 °C and supernatants were analyzed for CAB and M2CAB contents by LC-MS/MS. To quantify release kinetics, drug concentrations were measured in samples collected from the culture media from the same retention experiment using preloaded MDMs, at the same time when cells were harvested to quantify intracellular drug retention.

**Animal dosing and sampling.** Eight-week-old, healthy male Balb/cJ mice and Sprague-Dawley (SD) rats were purchased from Charles River Laboratories (Wilmington, MA). Both mice and rats were housed in the University of Nebraska Medical Center (UNMC) laboratory animal facility accredited by the American Animal Association and Laboratory Animal Care (AAALAC). Animals were housed under a 12-h light/dark cycle at a temperature of 20–24 °C and humidity range of 30–70%. The animals were maintained on sterilized 7012 Teklad diet (Harlan, Madison, WI), and water was provided ad libitum. All procedures were approved by the Institutional Animal Care and Use Committee (IACUC) at the University of Nebraska Medical Center (UNMC) as set forth by the National Institutes of Health (NIH).

Two treatment groups of a single intramuscular injection (IM, caudal thigh muscle; 1.35 µl/g and 1.23 µl/g of body weight for mice and rats, respectively) of 45 mg CAB equivalents/kg of NCAB or NM2CAB were used for both mice and rats. For each treatment of either NCAB or NM2CAB, four groups of mice and rats ($N = 6$) were dosed on day 0, and then at 1, 3, 6, and 12 months each group was sacrificed and tissues including liver, lymph nodes, kidneys, spleen, lungs, brain, gut, muscle, and muscle from the site of injection were collected. In mice, inguinal, axillary, branchial, and cervical lymph nodes were collected and pooled, while in rats only cervical lymph nodes were collected. Following injection, blood samples were collected at 4 h, days 1, 2, 4, 7, 14, 28, 42, and 56 and then monthly for 12 months into EDTA tubes (blood sample time points were evenly distributed between the four groups). Fifty µl of blood was immediately transferred into 950 µl methanol (containing 0.1% FA and 2.5 mM AF), vortexed, and stored at −80 °C until drug analysis. The remaining portion of the blood samples were centrifuged at $2000 \times g$ for 5 min for plasma collection, which was stored at −80 °C until drug analysis. A one-year PK study of NM2CAB, performed in male rhesus monkeys, was previously reported[15]. We report here PK assessment for samples collected for an additional year. Plasma samples were collected up to day 672 and CAB and M2CAB quantification were performed by LC-MS/MS. Metabolic chemistry panels were assessed in rat plasma on an Abaxis Vetscan VS2 chemistry analyzer (Zoetis Reference Laboratories, Parsippany-Troy Hills, NJ, USA) using a comprehensive diagnostic profile cartridge.

**Covance PK studies.** To provide added rigor and reproducibility to the data sets affirmation PK studies were performed by an independent contract laboratory. Male BALB/cAnNHsd (Balb/c) and female NOD.Cg-$Prkdc^{scid}$ $Il2rg^{tm1Wjl}$/SzJ (NSG) mouse studies were performed by Covance Laboratories (Greenfield, IN, USA). Briefly, 11-week-old male Balb/c and female NSG mice were purchased from Envigo RMS Inc. (Indianapolis, IN, USA) and Charles River Laboratories (Wilmington, MA, USA), respectively. Both mouse strains were housed at Covance Laboratories in an AAALAC-approved facility and the study was conducted using a Covance IACUC-approved animal protocol. Mice were housed under a 12-h light/dark cycle at a temperature of 20–26 °C and humidity range of 30–70%. Certified Rodent Diet #2014C (Envigo RMS, Inc.) was provided ad libitum to male animals and irradiated Rodent Diet #2920X (Envigo RMS, Inc.) was provided ad libitum to female animals. Male animals received Greenfield city water ad libitum and female animals received acidified water ad libitum. Animal treatments, clinical observations, sample collections, and sample analyses were conducted using standard operating procedures. Certified rodent diet #2014C (Envigo RMS, Inc.) was provided ad libitum to Balb/cJ mice and irradiated rodent diet #2920X (Envigo RMS, Inc.) was provided ad libitum to NSG mice. NCAB and NM2CAB formulations were prepared in the NNPP using GLP protocols. Formulation endotoxin levels were below 5 EU/kg. Formulations were packaged and shipped overnight at ambient temperature (18.1 °C average) to Covance Laboratories in Greenfield, IN. Drug concentrations in the formulations were determined by

UPLC-MS/MS at UNMC and by UPLC-UV/Vis at Covance. Personnel involved in drug administration, sample collection, and sample analysis were blinded as to the treatments. Animals were divided into eight treatment groups that received NCAB or NM2CAB ($N = 6$; subdivided into two groups of $N = 3$ for alternating bleeding time points) as a single IM injection (1.3 μl/g, caudal thigh muscle) of 45 or 70 mg CAB equivalents/kg. Blood samples were collected via submandibular puncture into tubes containing lithium heparin from day 14 and every other week thereafter, up to 183 days. These studies mirrored previously reported studies performed by the Gendelman laboratory at UNMC in male Balb/cJ and female NSG mice[15]. Plasma samples were stored at −80 °C until analysis. Frozen plasma aliquots from each animal at each time point were shipped on dry ice by overnight shipping to UNMC for drug analysis. Samples received at UNMC were stored at −80 °C until analysis.

**Sample preparation and LC-MS/MS analyses**. For blood samples, 50 μl blood collected in 950 μl of methanol was vortexed, centrifuged at 16,000 × g for 10 min at 4 °C, and 50 μl supernatant was aspirated, mixed with 50 μl IS [20 ng/ml d3-dolutegravir (d3-DTG) and 40 ng/ml stearoylated darunavir (SDRV)] in 70% methanol (containing 0.1% FA and 2.5 mM AF). d3-DTG was used as the IS (final concentration = 10 ng/ml) for the quantification of CAB, and SDRV (final concentration = 20 ng/ml) was used as the IS for quantification of M2CAB. Ten μl of the sample was analyzed by LC-MS/MS for CAB and M2CAB.

For tissue sample preparation, 3–150 mg of each tissue sample were homogenized in 4–35 volumes (depending on the tissue) of 80% methanol (containing 0.1% FA and 2.5 mM AF) using a TissueLyzer II (Qiagen, Valencia, CA, USA). Two hundred and ninety μl of methanol (containing 0.1% FA and 2.5 mM AF) and 10 μl 80% methanol was added to 100 μl of tissue homogenates. Samples were vortexed for 3 min, and centrifuged at 16,000 × g for 10 min. Then, 50 μl supernatant was aspirated and mixed with 50 μl IS in 80% methanol (containing 0.1% FA and 2.5 mM AF). Ten microliter sample was injected on LC-MS/MS for CAB and M2CAB analysis. Calibration curves in the range of 0.05–500 ng/ml for CAB and MCAB were prepared the same way in blank blood and tissues.

For LC-MS/MS quantification of CAB and M2CAB, a Waters ACQUITY UPLC system (Waters, Milford, MA, USA) connected to a Waters Xevo TQ-XS mass spectrometer with an electrospray ionization source was used. For CAB analysis, chromatographic separation was achieved on an ACQUITY UPLC BEH Shield RP18 column (2.1 × 100 mm, 1.7 μm; Waters) using 7-min gradient of mobile phase A (7.5 mM AF, pH 3) and mobile phase B (100% acetonitrile) at a flow rate of 0.25 ml/min. The initial mobile phase composition was 37% B for the first 4.0 min, increased to 95% B over 0.25 min, and held constant for 1.25 min. Mobile phase B was then reset to 37% over 0.5 min and the column was equilibrated for 1 min before the next injection. M2CAB chromatographic separation was achieved on ACQUITY UPLC BEH Shield RP18 column (2.1 × 30 mm, 1.7 μm; Waters) using 8-min gradient of mobile phase A (7.5 mM AF, pH 3) and mobile phase B (100% methanol), at a flow rate of 0.28 ml/min. The initial mobile phase composition was 85% B for the first 5 min, increased to 95% B over 0.25 min, and held constant for 1.5 min. Mobile phase B was then reset to 85% over 0.25 min and the column was equilibrated for 1 min before the next injection. CAB, M2CAB, d3-DTG, and SDRV were detected at a cone voltage of 2, 4, 2, and 70 V, respectively, and a collision energy of 22, 20, 16, and 16 V, respectively, in the positive ionization mode. Multiple reaction monitoring (MRM) transitions used for CAB, M2CAB, d3-DTG, and SDRV were 406.21 > 127.08, 672.47 > 406.16, 423.27 > 277.21, and 814.70 > 658.61, respectively. Spectra were analyzed and quantified by MassLynx V4.1 software. All calculations were made using analyte to IS peak area ratios.

**PK analyses**. Mean plasma drug concentrations were calculated per treatment group and PK parameters were derived using non-compartmental analysis of blood concentration vs. time profiles, using Phoenix WinNonlin V8.2 software. Peak plasma concentration ($C_{max}$), time to reach $C_{max}$ ($T_{max}$), elimination rate constant ($\lambda_Z$), half-life from elimination phase ($t_{0.5}$), area under the plasma concentration versus time curve (AUC), mean resident time (MRT), clearance (CL/F), and apparent volume of distribution ($V_z$/F) were calculated. Tissue concentrations were calculated and expressed as ng/g tissues.

**Statistical analyses**. For all studies, data were analyzed using Microsoft Excel V16.45 (Redmond, WA, USA) and GraphPad Prism V9.0.0.0 software (La Jolla, CA, USA) and presented as the mean ± the standard error of the mean (SEM). The number of replicates ($N$) is listed for each experiment. For comparing two groups, six animals/group will provide 80% power at the 0.05 level of significance to detect a difference of 2.0 standard deviations using a $t$-test. Extreme outliers beyond the 99% confidence interval of the mean and threefold greater than the SEM were excluded. No data were determined to fit this criterion, therefore no data points were excluded from the analysis. Significant differences were determined at $P < 0.05$.

**Study approvals**. All experimental protocols involving the use of laboratory animals were approved by the UNMC and Covance Institutional Animal Care and Use Committees (IACUC) ensuring the ethical care and use of laboratory animals in experimental research. All animal studies were performed in compliance with institutional policies and NIH guidelines for laboratory animal housing and care in American

Animal Association and Laboratory Animal Care (AAALAC) accredited facilities. Human monocytes were isolated by leukapheresis from HIV-1/2 and hepatitis B seronegative donors according to a UNMC Institutional Review Board (IRB) exempt protocol. All donors gave informed consent for the use of the deidentified material. Cells obtained from elutriation were negative for mycoplasma contamination.

**Reporting summary**. Further information on research design is available in the Nature Research Reporting Summary linked to this article.

## Data availability
The authors declare that the data supporting the findings of this study are available within the paper and its supplementary information files. Source data are provided at. https://doi.org/10.6084/m9.figshare.14496093 (Fig. 1) https://doi.org/10.6084/m9.figshare.14498025 (Fig. 2) https://doi.org/10.6084/m9.figshare.14498028 (Fig. 3) https://doi.org/10.6084/m9.figshare.14498034 (Fig. 4) https://doi.org/10.6084/m9.figshare.14498037 (Supplementary Fig. 1) https://doi.org/10.6084/m9.figshare.14498043 (Supplementary Fig. 2) https://doi.org/10.6084/m9.figshare.14498046 (Supplementary Fig. 3) https://doi.org/10.6084/m9.figshare.14498049 (Supplementary Fig. 4) https://doi.org/10.6084/m9.figshare.14498052 (Supplementary Fig. 5) https://doi.org/10.6084/m9.figshare.14498055 (Supplementary Fig. 6) https://doi.org/10.6084/m9.figshare.14498058 (Supplementary Fig. 7) https://doi.org/10.6084/m9.figshare.14519976 (Supplementary Table 1)

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

## Acknowledgements

We thank the University of Nebraska Medical Center (UNMC) Elutriation and Cell Separation Core (Myhanh Che and Na Ly) for providing human monocytes. We are thankful to Benjamin G. Lamberty, Brenda M. Morsey, and Dr. Howard Fox for the extension of previously published data sets performed with NM2CAB in rhesus macaques (in Nat Mater. 2020 Aug;19(8):910-920). The work was supported, in part, by the University of Nebraska Foundation, which includes donations from the Carol Swarts, M.D. Emerging Neuroscience Research Laboratory, the Margaret R. Larson Professorship, the Frances and Louie Blumkin, and the Harriet Singer Research Donations. We thank Dr. Bradley Britigan, Dean of the College of Medicine at UNMC, for providing funds to support the Covance studies. The authors thank Sarah Favara, Emily Archer and the Covance Laboratory staff that assisted in conducting and analyzing the data sets for rigor and reproducibility. We thank Dr. Jennifer Larsen, the Vice Chancellor for Research for UNMC for Core research support enabling continuance of this work. The research also received support from National Institutes of Health grants PO1 DA028555, R01 NS36126, PO1 MH64570, P30 MH062261, R01 AG043540, and 2R01 NS034239. We also thank the INBRE grant support from 2P20GM103427 for infrastructure research support.

## Author contributions

N.G. designed the study, executed the laboratory and animal PK experiments, performed the data analysis and interpretation, co-wrote the manuscript; J.M. assisted in study design, data analysis, and interpretation, co-designed and coordinated the Covance study and plasma sample analyses, edited the manuscript, and assisted in the preparation of the figures; D.K. performed the animal PK experiments, responsible for the data acquisition and analysis; A.B. assisted in animal study design and co-executed the animal PK experiments; Q.P. and W.L. assisted with the animal PK sample preparation and analysis and performed LC-MS/MS assays; T.K. assisted in human monocyte cell culture experiment and data analysis; B.S. synthesized the prodrug and prepared and characterized the nanoformulations for both the College of Pharmacy and Covance studies, assisted with LC-MS/MS tests of Covance mouse plasma samples, and edited the manuscript; N.S. assisted in animal PK sampling; B.L.D.S. performed LC-MS/MS analyses of all samples from the Covance study and assisted in data analysis from the study; A.S. prepared and characterized all formulations for the College of Pharmacy and Covance studies; B.E. assisted in study design, created nanoformulation, edited the manuscript, and co-funded the study; H.E.G. co-developed the study design, co-wrote the manuscript, developed the infrastructure responsible for Nebraska Nanomedicine Production Plant for the nanoformulations used in study, developed the Covance research design and provided funding support for the studies; Y.A. was responsible for the study design and data interpretation and co-wrote the manuscript.

## Competing interests

B.E. and H.E.G. are cofounders of Exavir Therapeutics, Inc. and are inventors on a patent that cover year-long integrase inhibitor prodrug formulations (PCT/US2019/057406, WO2020-086555). All other authors declare no competing interests.
