## [Peer Review File · Nature Communications]

REVIEWER COMMENTS

Reviewer #1 (Remarks to the Author):

Gautam et al. report detailed pharmacokinetic assessment of NM2CAB, a nanoformulated fatty acid ester prodrug of cabotegravir (CAB). The work builds on and substantially extends previous work showing ability of NM2CAB to result in sustained CAB levels above clinical benchmarks for at least one year, far exceeding the current injectable CAB LA formulation. The work is of high quality and comprehensive and supports advancement of NM2CAB for clinical evaluation. The work is timely given the anticipated approval of CAB LA for prevention.

The conclusions are well supported by data from mice, rat, and macaques. The authors elegantly elucidated the mechanisms behind the long persistence which was due to nanoparticle and not prodrug stability combined with slow release from depots in the injection site and tissues. No in vivo toxicity was seen. The manuscript is well written. I have only minor suggestions:

1- Lines 90-93. Consider rewriting this sentence ‘The early development of a year-long CAB, however, has met with a number of questions linked to reproducibility in different animal species and the underlying prodrug hydrolysis mechanisms to produce the observed extended drug half-life remained unclear’

2- Because the efficacy and PK of CAB LA was extensively studied in macaques and the PK scaling of CAB from macaques to humans is well established, the authors may wish in the Discussion to remind the reader of the inferred dose interval of NM2CAB in humans that sustains a plasma concentration above $4 \times C_{90}$. Also highlighting advantages in injection volume, and formulation manufacture are equally relevant.

Reviewer #2 (Remarks to the Author):

In the manuscript “Prodrug-Release Defines the Extended Pharmacokinetic Profiles of a Year-long Cabotegravir”, authors present additional data to previous publications, namely, Kulkarni, T.A. et al. *Nat. Mater* (2020) and Zhou, T et al. *Biomaterials* (2018). Previous publications described development of modified cabotegravir NM2CAB, that seems to have extended release compared to cabotegravir, an HIV antiretroviral drug currently in clinical trial. In particular, this manuscript repeats PK analysis of NM2CAB in 2 mouse strains (one same as in *Nat. Mater* (2020) and one new) in different setting (contract research laboratory). PK data in rats and later timepoints of PK in

macaques, that were not available in time of publication of previous manuscript are shown. Additional characteristics for stability studies were added. Prodrug distribution in tissues were also analyses in previous publications, in this manuscript, stability data from a second mouse strain and rat are presented. Overall, the manuscript does not describes new observations compared to previously published studies by same group.

Additional comments:

1. In the abstract it should be clearly stated which animal model was use for data. For example, it is true that plasma level is above IC90 more than year in rodents, but not for NHP.
2. The efficacy macaque studies with the long acting cabotegravir currently in clinical trials identified plasma drug concentrations more than 4 times the protein-adjusted 90% inhibitory concentration ($4 \times \text{PA-IC}_{90}$) as a correlate of HIV protection (Andrews CD et al., *Sci. Transl. Med* 2015, Radzio J., *Sci. Transl. Med* 2015, Andrews CD et al., *AIDA*, 2017), Dobard C, et al., *J. Infect. Dis* 2020). PK studies of the formulation in this manuscript does not reach that level. Without efficacy data, it seems that higher dose will be needed. Please, clarify choice of dose for experiments in this manuscript and relevance to potential translation to human.
3. Abstract states that “no toxicities were recorded”, however the manuscript does not show any toxicity studies.
4. Stability assessed for 1 year as release kinetic was in multiple timepoints >100%. Please clarify how stability can improve with time and what is the reference point.
5. Chemical stability is not measured during storage, rather it refers to stability of solubilized form of the prodrug in different pH. How this relates to stability of the formulation during storage?
6. Biological stability is referring to chemical stability of prodrug in blood, S9 liver fraction and tissue homogenates of various species, rather than to biological activity of the formulation, i.e. inhibition of HIV integrase. This should be clarified.
7. Figure 5 refers to : “Pharmacokinetic and biodistribution models of NM2CAB”. However, the manuscript does not describe any modeling. This is rather cartoon. It is also misleading as it shows human figures, but tissue distribution was analyzed in mouse and rats. No human sample was tested. These cartoons should be changed or removed.

March 5, 2021

Thank you for the thorough and reasoned review of our manuscript titled, "Lipophilic Nanocrystal Prodrug-Release Defines the Extended Pharmacokinetic Profiles of a Year-Long Cabotegravir". We are pleased to respond to each of the queries point-by-point. The amendments made in the text **are highlighted in yellow.**

Reviewer 1

Overall. *"Gautam et al. report detailed pharmacokinetic assessment of NM2CAB, a nanoformulated fatty acid ester prodrug of cabotegravir (CAB). The work builds on and substantially extends previous work showing ability of NM2CAB to result in sustained CAB levels above clinical benchmarks for at least one year, far exceeding the current injectable CAB LA formulation. The work is of high quality and comprehensive and supports advancement of NM2CAB for clinical evaluation. The work is timely given the anticipated approval of CAB LA for prevention. The conclusions are well supported by data from mice, rat, and macaques. The authors elegantly elucidated the mechanisms behind the long persistence which was due to nanoparticle and not prodrug stability combined with slow release from depots in the injection site and tissues. No in vivo toxicity was seen. The manuscript is well written. I have only minor suggestions"*

Response. Thank you.

Point 1. *"Lines 90-93. Consider rewriting this sentence 'The early development of a year-long CAB, however, has met with a number of questions linked to reproducibility in different animal species and the underlying prodrug hydrolysis mechanisms to produce the observed Page 5 of 6 extended drug half-life remained unclear"*

Response: The sentence was rewritten to best clarify the study's purpose. *"However, during the development of a year-long CAB reproducibility of PK data sets in different animal species proved mandatory for development as was unraveling the mechanism(s) of the extended drug apparent half life that included prodrug hydrolysis. Moreover, safety measurements for the year-long parenteral formulation were also required."*

Point 2. *"Because the efficacy and PK of CAB LA was extensively studied in macaques and the PK scaling of CAB from macaques to humans is well established, the authors may wish in the Discussion to remind the reader of the inferred dose interval of NM2CAB in humans that sustains a plasma concentration above 4xIC₉₀. Also highlighting advantages in injection volume, and formulation manufacture are equally relevant"*

Response: The discussion was modified to include inferred dose intervals, lowered injection volume and ease of formulation manufacture required to sustain a plasma dose above 4xIC₉₀. Briefly, when NM2CAB is administered for pre-exposure prophylaxis the dose required is 1xIC₉₀. Notably, plasma concentrations of NM2CAB were up to, or greater than, 100-times those resulting from the equivalent NCAB (a CAB LA equivalent formulation) for time periods of a year. The other advantages of NM2CAB for long-acting treatment are several-fold. *First*, NM2CAB can be manufactured at drug concentration of 400 mg/ml, twice the CAB LA concentration of 200 mg/ml. Thus, NM2CAB would require a 1 ml injection volume for a yearly delivery in comparison to the 2 ml monthly injection for CAB LA. *Second*, for the NM2CAB formulations developed no injection site reactions were observed in any animal species. Thus NM2CAB would provide an improvement over the current approved CAB LA

formulation with respect to extended dosing interval and reduced injection volume and no injection site reactions.

Reviewer 2

Overall. *“In the manuscript “Prodrug-Release Defines the Extended Pharmacokinetic Profiles of a Year-long Cabotegravir”, authors present additional data to previous publications, namely, Kulkarni, T.A. et al. Nat. Mater (2020) and Zhou, T et al. Biomaterials (2018). Previous publications described development of modified cabotegravir NM2CAB, that seems to have extended release compared to cabotegravir, an HIV antiretroviral drug currently in clinical trial. In particular, this manuscript repeats PK analysis of NM2CAB in 2 mouse strains (one same as in Nat. Mater (2020) and one new) in different setting (contract research laboratory). PK data in rats and later timepoints of PK in macaques, that were not available in time of publication of previous manuscript are shown. Additional characteristics for stability studies were added. Prodrug distribution in tissues were also analysed in previous publications, in this manuscript, stability data from a second mouse strain and rat are presented. Overall, the manuscript does not describe new observations compared to previously published studies by same group.”*

Response: We respectfully disagree on this point. The manuscript does provide new comprehensive observations for the rigor and reproducibility of the prodrug formulation pharmacokinetics in different species. Also, the long apparent drug half-life was found herein to follow pH-dependent prodrug hydrolysis upon slow dissolution of the nanocrystal. In contrast, solubilized prodrug was hydrolyzed in hours in plasma and tissues recorded from multiple mammalian species at basic pH. This is a few mechanisms for how prodrugs can extend the apparent half life of their native counterparts and would be applicable to other agent classes. Importantly the results, taken together, serve to affirm the pharmacological properties of a year-long nanoformulated CAB prodrug supporting the novel mechanistic design for drug formulation safety, rigor and reproducibility. This is critically important as a safe, year-long agent that could prevent HIV transmission could have a substantive impact on transmission of the virus and hence on the pandemic.

Point 1. *“In the abstract it should be clearly stated which animal model was used for data. For example, it is true that plasma level is above IC90 more than year in rodents, but not for NHP.”*

Response: The data obtained from the animal model used in the abstract is now clearly stated.

Point 2. *“The efficacy macaque studies with the long acting cabotegravir currently in clinical trials identified plasma drug concentrations more than 4 Times the protein-adjusted 90% inhibitory concentration (4xPA IC90) as a correlate of HIV protection (Andrews CD et al., Sci. Transl. Med 2015, Radzio J., Sci. Transl. Med 2015, Andrews CD et al., AIDA, 2017), Dobard C, et al., J. Infect. Dis 2020). PK studies of the formulation in this manuscript does not reach that level. Without efficacy data, it seems that higher dose will be needed.*

Please, clarify choice of dose for experiments in this manuscript and relevance to potential translation to human.”

Response. The reported monkey data in this manuscript is an extension of our prior report (Nat Mater. 2020 Aug;19(8):910-920. doi: 10.1038/s41563-020-0674-z). We achieved an elimination half-life of 180 days. The dosing concentration was used for injection reflected what was first used in the rodents. Based on the release mechanism observed in the prior and in this report, we agree with the reviewer that further dose optimizations are needed for volume and drug concentrations to achieve levels above the 4xIC90 for a year after a single injection. Nonetheless, the levels achieved herein are at or above 1xIC90 and are sufficient to prevent viral transmission for what was the relevant translational end point in the current report. While it is true that a higher dose is needed for treatment and prolonged viral suppression we have clarified the dose choice for these experiments and the relevance to potential human translation in the discussion. Also as noted in a review article by Andrews (<https://www.natap.org/2015/HIV/Cabotegravir.pdf#ion.10.pdf>), the half

life of CAB LA in macaques is shorter (8-9 days) compared to humans (21-50 days), requiring more frequent dosing (50 mg/kg) in the referenced macaque PrEP studies. For our study, a dose of 45 mg/kg was used which provides opportunities for future studies where the dose concentrations would be expanded.

Point 3. “Abstract states that “no toxicities were recorded”, however the manuscript does not show any toxicity studies.”

Response: We have extensively explored potential toxicities for these novel formulations. Neutrophil, lymphocyte and monocyte counts and serum metabolic profiles through day 672 following NM2CAB treatment of the NHPs were within normal limits. Animal weights were equivalent to untreated controls. These data are shown below the controls (standard controls).

Point 4. “Stability assessed for 1 year as release kinetic was in multiple timepoints >100%. Please clarify how stability can improve with time and what is the reference point.”

Response: Stability was calculated as the ratio of total drug present at all time points against day 0. Data >100% was due to analytical errors linked to individual sample measurement. Our bioanalytical method passed validation by US-FDA guidelines allowing for ±15% accuracy.

<https://www.fda.gov/files/drugs/published/Bioanalytical-Method-Validation-Guidance-for-Industry.pdf>

Point 5. “Chemical stability is not measured during storage, rather it refers to stability of solubilized form of the prodrug in different pH. How this relates to stability of the formulation during storage?.”

Response. Formulations in the undiluted and 20x diluted form were stored for a year. At multiple time points, samples were collected and total drug concentration were measured. Total drug content versus day 0 reflects stability, whereas drug content in supernatant fluids against levels at day 0 reflects release. Therefore, both storage stability and release kinetics were monitored in a single experiment.

Point 6. *“Biological stability is referring to chemical stability of prodrug in blood, S9 liver fraction and tissue homogenates of various species, rather than to biological activity of the formulation, i.e. inhibition of HIV integrase. This should be clarified..”*

Response. We previously referred to stability in biological matrices as biological stability, but we agree with reviewer that this may be confused with the biological activity against HIV. Therefore, we have replaced the term “biological stability” with “metabolic stability” throughout the manuscript. This point is now clarified.

Point 7. *“Figure 5 refers to : “Pharmacokinetic and biodistribution models of NM2CAB”. However, the manuscript does not describe any modeling. This is rather cartoon. It is also misleading as it shows human figures, but tissue distribution was analyzed in mouse and rats. No human sample was tested. These cartoons should be changed or removed.”*

Response. The cartoon was removed. We agree with the reviewer.

We wish to thank you and the reviewers for your thoughtful comments. We look forward to the publication of this work. Please do not hesitate to call on me once again if any further needs are required.

With best regards,

Howard E. Gendelman, M.D.

Margaret R. Larson Professor of Internal Medicine and Infectious Diseases

Professor and Chairman, Department of Pharmacology and Experimental Neuroscience

REVIEWER COMMENTS

Reviewer #1 (Remarks to the Author):

The authors may wish to additionally discuss the dose optimization needed to achieve 4xPA-IC90 in the Rhesus macaques. Recent data presented at CROI on the HPTN 083 CAB LA among MSM confirmed that rare breakthrough infections do occur at the 4xPA-IC90 target further supporting that the clinical target exposure for prevention will be at least 4XPA-IC90 and not 1XPA-IC90 or other lower concentrations. The manuscript will benefit from a discussion acknowledging this target and exploring potential dose optimization in Rhesus macaques and injection volumes that would achieve this plasma concentration.

Reviewer #2 (Remarks to the Author):

During original review of this manuscript this reviewer notice no new observations compared to previously published papers by the same group. Unfortunately, this resubmission of the manuscript did not change this view.

Authors obviously disagree with this point and claim that “the manuscript does provide new comprehensive observations for the rigor and reproducibility of the prodrug formulation pharmacokinetics in different species”. While rigor ensures robust and unbiased experimental design and methodology, reproducibility warrants that results can be reproduced by multiple scientists. Indeed, authors were able to reproduce their already published data. Rigor and reproducibility however are expected in all published work especially in high quality journal like Nature Mater., where this work was published last year. With exception of additional strain of mice, no new species was used compared to previous publication. Authors claim that they now described the mechanism of extended release that involved hydrolysis of the prodrug. However, experiment with fast prodrug analysis were also shown in previous publication.

Several points were not addressed satisfactory.

Point 2: Several publications showed that plasma concentration of CAB for HIV prevention in NHP needs to be 4xIC90. These studies were not suggesting 4xIC90 for treatment, but for HIV transmission. PK in the manuscript, however, showed plasma level only above IC90. Authors were asked to clarify choice of dose for the experiments and discuss possible translation for human. This request was not addressed in the new version of the manuscript.

Point 3: Abstract is still claiming “no toxicities were recorded” while no data are presented in the manuscript. Instead, authors sent part of Figure 6 from already published paper in Nature Mater. This further underline criticism on lack of novelty of this manuscript. Please, for this manuscript provide new unpublished toxicity data or remove the statement from the abstract.

Point 4: Authors claim that data >100% in the Table 2 were due to analytical errors and that the method they are using has $\pm 15\%$ accuracy. It is correct that US-FDA guidelines allows max $\pm 15\%$. However, with this accuracy it is impossible to find “8% release of M2CAB from NM2CAB at day 0” as 8% is lower than accuracy of the method. Please, address this issue in the manuscript. Also, statement about accuracy of the method used have to be clearly stated in the method section and in the table 2 as this is important to explain of data >100%.

Point 5: Chemical stability ensure that a chemical in a formulation is the same at the beginning and in the end of the test. Data in the manuscript does not show identity of the drug in formulation and/or amount of drug in formulation before and after stability test. Instead, release of prodrug from nanoformulation and fast hydrolysis of the released prodrug (lines 210-216) are shown. This is not chemical stability. No explanation is provided why this is call chemical stability. How is this relevant as only 8% of the drug is released?

April 5, 2021

Re: NCOMMS-21-02965A "Lipophilic Nanocrystal Prodrug-Release Defines the Extended Pharmacokinetic Profiles of a Year-Long Cabotegravir"

Thank you for the thorough and reasoned review of our manuscript titled, "Lipophilic Nanocrystal Prodrug-Release Defines the Extended Pharmacokinetic Profiles of a Year-Long Cabotegravir". We are pleased to respond to each of the referees queries point-by-point. The amendments made in the text are highlighted in yellow.

Reviewer 1

Point 1. "The authors may wish to additionally discuss the dose optimization needed to achieve 4xPA-IC90 in the Rhesus macaques. Recent data presented at CROI on the HPTN 083 CAB LA among MSM confirmed that rare breakthrough infections do occur at the 4xPA-IC90 target further supporting that the clinical target exposure for prevention will be at least 4XPA-IC90 and not 1XPA-IC90 or other lower concentrations. The manuscript will benefit from a discussion acknowledging this target and exploring potential dose optimization in Rhesus macaques and injection volumes that would achieve this plasma concentration."

Response: We have added to our discussion dose optimization needed to achieve 4xPA-IC90 in nonhuman primates (NHP). This includes both dose escalation and dosing volumes taking into consideration a 400 mg/ml drug concentration. We also understand the needs to achieve a higher plasma CAB dose over the yearlong interval and the need to explore potential dose optimization and injection volumes. To this end we have outlined below and placed into the modified text of discussion the steps that will be needed to reach 4XPA-IC90 for NM2CAB based on our current and future preclinical pharmacokinetics (PK). *First*, we must extend analyses of the CAB M2CAB prodrug and its CAB metabolite in dose escalating studies performed *in vivo* in each of our mice, rat, and monkey models. *Second* each all of our preclinical tissue distribution and drug excretion need be investigated. These will include, but not be limited to, plasma protein binding and hepatic microsomal metabolic stability. The (AUC0-inf) and C30s of NM2CAB at 5 to 500 mg/kg and the C2min of NM2CAB will determine pharmacokinetic profiles obtained at varying dose. As the NM2CAB prodrug, can be quickly hydrolyzed into CAB and is protected by the nanofomulation after intramuscular administration the mechanisms seen in this report will serve as the foundation for these studies. *Third*, as NM2CAB easily-readily penetrates lymphoid tissue the concentrations in gut, lymph nodes, spleen and brain at each of these doses need be considered. While we understand that NM2CAB is a promising candidate prodrug required for clinical development we are pleased to report that a recent National Institutes of Health submitted grant from our group to perform such studies scored a three percentile (titled LASER ART for PreP, 1 R01 AI58160-01) to exactly do this work. We anticipate that such studies can be completed within the next two to three years before moving forward to early phase clinical studies.

Reviewer 2

Overall. "During original review of this manuscript this reviewer notice no new observations compared to previously published papers by the same group. Unfortunately, this resubmission of the manuscript did not change this view. Authors obviously disagree with this point and claim that "the manuscript does provide new comprehensive observations for the rigor and reproducibility of the prodrug formulation pharmacokinetics in different species". While rigor ensures robust and unbiased experimental design and methodology, reproducibility warrants that results can be reproduced by multiple scientists. Indeed, authors were able to

reproduce their already published data. Rigor and reproducibility however are expected in all published work especially in high quality journal like Nature Mater., where this work was published last year. With exception of additional strain of mice, no new species was used compared to previous publication. Authors claim that they now described the mechanism of extended release that involved hydrolysis of the prodrug. However, experiment with fast prodrug analysis were also shown in previous publication.” Several points were not addressed satisfactory.

Response: We are pleased to affirm the new observations made in this manuscript compared to any previously published papers by our own group and those of others. Clarity is provided in a reasoned and complete manner. *First*, in regard to rigor and reproducibility of these novel prodrug formulation PK we provide robust and unbiased experiential mythology reproduced by multiple scientists. These PK studies were performed independently in the laboratory of Dr. Yazen Alnouti. Dr. Alnouti was provided the prodrug formulations and asked to rigorously evaluate the study data sets and this analysis was done independently. In the prior works (**Nature Materials** 2020 19(8) pages 910-920) each and all the PK data sets were performed in separate laboratories that includes Drs. Howard Gendelman and Benson Edagwa (rodent studies) and Howard Fox (non-human primates). While Drs. Yazen Alnouti and Nagsen Gautam were co-authors on the prior study their engagements were restricted to the development of analytical methods for the cabotegravir (CAB) native and prodrug levels in plasma and tissue samples and PK analysis. These analyses employed a Waters ACQUITY UPLC system (Waters, Milford, MA, USA) connected to a Waters Xevo TQ-S-micro mass spectrometer for quantification of CAB and the CAB prodrug that was created (M2CAB). While each of the authors reviewed each and all of the data sets performed by others the work had not been repeated in any other laboratory. Thus, to ensure robust and unbiased experimental design and methodology the current work was done independently by Drs. Gautam and Alnouti at all stages of testing and data analysis. *Second*, there were notably different species tested. These species included Sprague Dawley outbred rats and NSG immune deficient mice. Moreover, there were mixtures of male and female animals used. *Third*, we had taken the issue of rigor and reproducibility yet another step forward by employing a contract laboratory (Covance Laboratories, a global contract research organization and drug development services company) to repeat the study findings using multiple mouse strains (immune deficient and competent). *Fourth*, the analytical analyses were then repeated from samples obtained by the Covance Laboratories at the University of Nebraska Medical Center (UNMC). So as far the analytical data sets analyses were performed at two independent UNMC laboratories with two independent LC-MS/MS systems by two separate groups of scientists. Not to be limited each and all of these analyses was done yet again and a third time by the Covance Laboratories. We have outlined this process in the revised methods section of the manuscript. *Fifth*, I might add the reason for performing all of these new experiments for rigor and reproducibility. This was as a direct result of two factors. As a consequence of a group investigation that I myself participated in on the transparent reporting used to optimize the predictive value of preclinical research [please see **Nature** 2012 volume 490, pages187–19] changes in how experiments are performed were made including new strategic additions to all NIH grant applications. The highly impactful report came as a consequence of the “*US National Institute of Neurological Disorders and Stroke initiative where a convened conference was made with major stakeholders in June 2012 to discuss how to improve the methodological reporting of animal studies in grant applications and publications. The main recommendations made is that at a minimum study should report on sample-size estimation, whether and how animals were randomized, whether investigators were blind to the treatment, and the handling of data.*” As we would agree that “the dissemination of knowledge is the engine that drive scientific progress” and the claims offered in this study are a paradigm shift in knowledge and potential clinical application we believed this to be a required independent study. This is exactly what was performed as a consequence. While we would agree that *rigor and reproducibility is expected in all published works there are a multitude of examples* of reported studies that are well-designed and well-conducted but analysis that reporting limitations were found to correlate with overstated findings [**Nature** 2012 volume 490, pages187–19]. A yearlong antiretroviral drug when developed for human use could lead to impactful results in preventing or treating HIV/AIDS and ensuring that the data sets reported are at a higher level of accuracy is not only needed but expected. We contend that such analyses set a high but necessary bar for rigor and reproducibility in animal studies. Other major differences between the current study and prior works include the following

outlined below and are reflective of determining the **mechanism for how the drug's apparent half-life can be extended from a month to a year**. The critical new steps made in the current study are outlined as follows.

- (1) We showed that our nanoformulated prodrug of cabotegravir (coined NM2CAB) is retained at the injection site and in the reticuloendothelial system where it serves as depots that slowly release the prodrug.
- (2) We demonstrated that M2CAB is slowly released from NM2CAB, which then quickly undergoes enzymatic and non-enzymatic hydrolysis into the parent CAB that is detected in blood and tissues
- (3) We provided a linkage between the observed *in vivo* PK data sets to our *human monocyte-derived macrophage* uptake, retention and release. These data sets correlated with each of the chemical and biological nanoformulated prodrug stability recordings.
- (4) We extended prior analyses of formulation stability. Indeed, our prior studies assayed M2CAB alone and these studies were done in plasma to simply compare hydrolysis pattern in different species. To extend these works and to determine precise mechanisms we used whole blood for our analyses. What was now tested and beyond what had been previously reported included the M2CAB prodrug alone, its nanoformulation NM2CAB, a control NCAB (reflecting what has been currently approved for human use), and the parent CAB. These comparisons were required to elucidate the mechanism of the extended drug apparent half-life which was not determined in any prior report. *monocyte-derived macrophage* uptake, retention and release. These data sets correlated with each of the chemical and biological nanoformulated prodrug stability recordings.
- (5) Assay of tissue homogenate, liver S9 fractions, cell lysates, culture media and escalating pHs provided new insights into formulation and prodrug stability. Each and all were used to elucidate stability profiles in the different matrices and not limited to plasma.
- (6) New studies of particle uptake and retention were provided beyond analyses of NM2CAB and NCAB. In the current investigation NM2CAB, NCAB, M2CAB, and CAB, were used. This proved necessary in order to explain how the nanoformulation itself was able to protect the prodrug from rapid hydrolysis demonstrating the uptake, retention and clearly demonstrating the release advantages of NM2CAB compared to CAB, NCAB or M2CAB.
- (7) Storage and release kinetics were addressed. Here we demonstrate the significance of keeping hydrolyzable compounds in solid state within an aqueous formulation suspension during storage. This was done to affect drug stability and release kinetics. These observations are critical to product development and its future clinical use.
- (8) UNMC College of Pharmacy laboratory designed, implemented, performed and analyzed the PK studies in Balb/c mice and male Sprague Dawley rats. They independently performed dosing and sampling. Covance Laboratories (Greenfield, IN, USA), an independent contract laboratory provided added rigor and reproducibility to these studies as well as performing dose escalations.

Point 1. *“Several publications showed that plasma concentration of CAB for HIV prevention in NHP needs to be 4xIC90. These studies were not suggesting 4xIC90 for treatment, but for HIV transmission. PK in the manuscript, however, showed plasma level only above IC90. Authors were asked to clarify choice of dose for the experiments and discuss possible translation for human. This request was not addressed in the new version of the manuscript.”*

Response: We are aware of the prior studies demonstrating plasma CAB concentration required for simian immunodeficiency virus (SIV) prevention. These prior studies are reviewed in (**Expert Opin Investig Drugs** 2018 27(4):413-420; **Curr Opin HIV AIDS** 2015 10(4) 258-63). Such prior works including what was reported in **AIDS** 2017 31(4):461-467 demonstrate that the terminal phase half life of CAB in macaques is shorter (3-12 days) compared to humans (21-50 days). Drug administration requires sequential dosing at 50 mg/kg in the rhesus macaque for pre-exposure prophylaxis (PrEP) with challenge at week 2. In these studies, doses

of 10 mg/kg and 30 mg/kg failed to provide CAB levels above 4× IC90. Notably, using dose extrapolation, for a 70 kg person, a human equivalent dose of 50 mg/kg in rhesus macaques used for PrEP studies translates to 1129 mg of CAB LA in humans (**J Basic Clin Pharm** 2016 7(2):27-31). This is notably higher than the 400 and 600 mg that has now been extensively characterized in humans for monthly or bimonthly dosing and shown to be effective. These data underscore interspecies differences in CAB metabolism. Moreover, the variabilities in reported drug concentrations for NHP are also notable for CAB LA. For our study, a lower dose of 45 mg/kg was used and therefore dose ranging studies will be evaluated in our future studies. At the outset of this work, we did not know how to predict the therapeutic dose from our prodrug nanoformulations. However, our NM2CAB PK data in mice revealed dose-dependent increase in CAB suggesting that dose adjustment will be needed for future rhesus macaque experiments. Thus, the variabilities in reported drug concentrations for NHPs are reflected by, for example, a human dose of 600 mg CAB/70 kg which is equivalent to 8.5 mg/kg. This is followed by sequential dosing at 400 mg/70 kg and equal to 5.7 mg/kg monthly maintenance. Moreover, for “*Estimating the Maximum Safe Starting Dose in Initial Clinical Trials for Therapeutics in Adult Healthy Volunteers*” US Food and Drug administration (FDA) guidelines (<https://www.fda.gov/media/72309/download>) demonstrate scaling factors from human to mice at 12.3, rats at 6.2, and NHP at 3.1. This then translates to mice, rats and nonhuman primates of doses of 104.5, 52.8 and 26.3 mg/kg, respectively. Taken all of this together, the prediction of the maximum recommended therapeutic dose (MRTD) for antiretroviral prodrug nanoformulations and extrapolations on drug absorption, permeability, hydrolysis and apparent half life will be carefully calculated in the next stage of development. We also do not yet know whether or not what the dosing will be required to prevent SIV transmission in NHP with NM2CAB. We posit that it is likely that the dose for NM2CAB will be lower than what is required for CAB LA based on the following: (1) drug penetrance into lymphoid reservoirs of virus is more than a log greater between NM2CAB and CAB LA; (2) that the PK profiles show very limited fluctuation swings between NM2CAB compared to CAB LA; and (3) limited dosing volumes with formulations of 400 mg/ml compared against 200 mg/ml injection volumes. However, and as stated dose escalation and preexposure prophylaxis experimental testing will be required to affirm any of these hypotheses. This will all be done in future works (please see responses to reviewer 1 above). So, the work will move forward, and these questions will be addressed. The reported NHP data in this manuscript is an extension of our prior report (**Nature Materials** 2020 19(8):910-920). Based on the release mechanism observed in the prior and in this report, we agree with the reviewer that further dose optimizations are needed for volume and drug concentrations to achieve levels above the 4xIC90 for a year after a single injection (see also response to reviewer 1).

Point 2. *“Abstract is still claiming “no toxicities were recorded “while no data are presented in the manuscript. Instead, authors sent part of Figure 6 from already published paper in Nature Mater. This further underline criticism on lack of novelty of this manuscript. Please, for this manuscript provide new unpublished toxicity data or remove the statement from the abstract.”*

Response: The following safety data was placed in the supplemental files for both Balb/c mice and Sprague Dawley rats. This included measure of animal well-being and weights and parallel recordings of metabolic, liver and renal profiles. As per the former in mice, initial weight at dosing time was 18-23 g, at the study end at one year was 28-35 g and included no differences between treatments and controls. In rats, initial weight at dosing time was 186-223 g, by the study end at one year was 500-562 g. There were no deaths observed in any of the animal groups including samplings made and as a result of anesthesia. In regard to visual signs of local toxicities. No toxicities were observed at site of injection. For the latter comprehensive serum chemistry and tissue histopathology profiles were performed and were normal. Nonhuman primates were monitored up to two years with no evidence of any adverse reactions. Former studies provided one year only toxicological measures.

Point 3. *Authors claim that data >100% in the Table 2 were due to analytical errors and that the method they are using has ±15% accuracy. It is correct that US-FDA guidelines allows max ±15%. However, with this accuracy it is impossible to find “8% release of M2CAB from NM2CAB at day 0” as 8% is lower than accuracy of the method. Please, address this issue in the manuscript. Also, statement about accuracy of the method*

used have to be clearly stated in the method section and in the table 2 as this is important to explain of data >100%.

Response: Each of the analytical methods used in the current manuscript were validated according to the standard US FDA guidelines and passed all validation criteria including at least 15% accuracy and precision. Therefore, any reported concentration value is $\pm 15\%$ accurate. For example, % stability is equal to measured total drug concentration at each of the defined time points divided by the original drug concentration at day 0. Therefore, both the time points as well as day 0 measured concentrations are within $\pm 15\%$ accurate. For the ratio of the two concentrations, i.e., % stability, accuracy of this number is determined by the accuracy of both concentration components. It is common that when a drug is stable, one can record % stability >100%. Some report such data as 100%, but we chose to report the numbers as is for transparency. The same exact argument applies for % release.

Point 4. *“Chemical stability ensures that a chemical in a formulation is the same at the beginning and in the end of the test. Data in the manuscript does not show identity of the drug in formulation and/or amount of drug in formulation before and after stability test. Instead, release of prodrug from nanoformulation and fast hydrolysis of the released prodrug (lines 210-216) are shown. This is not chemical stability. No explanation is provided why this is call chemical stability. How is this relevant as only 8% of the drug is released?”*

Response: Storage stability and release kinetics were determined for both NM2CAB and NCAB. These are as storage stability (combined released and unreleased total drug content in mass balance vials) and released drug concentrations in the supernatant after pelleting the intact nanoparticles. This mixture contained unreleased drug. The data showed that M2CAB content was 100 % and this was recovered from all time points in the mass balance set that was shown to be 100% stable. In addition, equivalent results were seen with the recording of >90% M2CAB unreleased within the nanoparticles and reflecting minimal drug release. We have not observed conversion of M2CAB to CAB during storage in solid form. The solid state of the hydrophobic (aqueous-0.10 $\mu\text{g}/\text{mL}$) and lipophilic (octanol-8057.2 $\mu\text{g}/\text{mL}$) M2CAB nanocrystals contributes towards prodrug chemical stability within the aqueous formulation. However, in a separate experiment we determined the chemical stability of the unformulated and solubilized M2CAB at different pHs. We found that the prodrug was unstable due to rapid chemical hydrolysis into parent CAB. Combining the data sets provides support of the up to 100% stability of M2CAB solid drug particles within NM2CAB nanoformulation. Based on this data we posit that M2CAB is protected from chemical degradation when its in solid state within surfactant stabilized aqueous nanoparticles. This is divergent from the chemical stability of M2CAB solution. The prodrug itself when separated from the surfactant coating and in solution form is unstable. Taken together the solid state of M2CAB within surfactant stabilized aqueous nanosuspension contributes to chemical prodrug stability. We are pleased to affirm the PK properties of a year-long nanoformulated CAB prodrug and further demonstrating novel mechanistic design for drug formulation safety, rigor and reproducibility. This is critically important as a safe, year-long agent that could prevent HIV transmission could have a substantive impact on transmission of the virus and hence on the pandemic. The interest in yearlong acting antiretrovirals is undeniable from both prevention and eradication paradigms. We wish to thank you and the reviewers once again for your thoughtful comments. We look forward to the publication of this work. Please do not hesitate to call on me once again if any further needs are required.

With best regards,

Howard E. Gendelman, M.D.

Margaret R. Larson Professor of Internal Medicine and Infectious Diseases
Professor and Chairman, Department of Pharmacology and Experimental Neuroscience

REVIEWERS' COMMENTS

Reviewer #1 (Remarks to the Author):

The authors response to my comment is satisfactory.

Thank you

Walid Heneine

Reviewer #2 (Remarks to the Author):

This reviewer appreciates that in this revision, authors satisfactory addressed all my comments.